

# Maniraptoran pelvic musculature highlights evolutionary patterns in theropod locomotion on the line to birds

Matthew M. Rhodes[1], Donald M. Henderson[2] and Philip J. Currie[1]

[1] Department of Biological Sciences, University of Alberta, Edmonton, Alberta, Canada
[2] Royal Tyrrell Museum of Palaeontology, Drumheller, Alberta, Canada

## ABSTRACT

Locomotion is a fundamental aspect of palaeobiology and often investigated by comparing osteological structures and proportions. Previous studies document a stepwise accumulation of avian-like features in theropod dinosaurs that accelerates in the clade Maniraptora. However, the soft tissues that influenced the skeleton offer another perspective on locomotory adaptations. Examination of the pelvis for osteological correlates of hind limb and tail musculature allowed reconstruction of primary locomotory muscles across theropods and their closest extant relatives. Additionally, the areas of pelvic muscle origins were quantified to measure relative differences within and between taxa, to compare morphological features associated with cursoriality, and offer insight into the evolution of locomotor modules. Locomotory inferences based on myology often corroborate those based on osteology, although they occasionally conflict and indicate greater complexity than previously appreciated. Maniraptoran pelvic musculature underscores previous studies noting the multifaceted nature of cursoriality and suggests that a more punctuated step in caudal decoupling occurred at or near the base of Maniraptora.

## INTRODUCTION

Locomotion is a vital component of palaeobiology as it determines how individuals acquire resources and escape predators, and is inherently linked to other aspects such as body size and physiology (*Hutchinson, 2004a*; *Hutchinson, 2004b*). Of particular interest is locomotion in theropod dinosaurs, whose fossil record documents the gradual but steady assembly of the avian body plan (*Hutchinson & Allen, 2009*; *Brusatte et al., 2014*; *Xu et al., 2014*). Non-avian theropods achieved global distribution and striking diversity while experimenting with locomotion at large body sizes encroaching on biomechanical limits for bipeds (*Hutchinson & Garcia, 2002*; *Henderson & Snively, 2004*; *Therrien & Henderson, 2007*; *Hutchinson et al., 2011*; *Persons & Currie, 2016*; *Persons, Currie & Erickson, 2020*; *Dececchi et al., 2020*), flight and wing-based locomotion (*Burgers & Chiappe, 1999*; *Chatterjee & Templin, 2007*; *Alexander et al., 2010*; *Evangelista et al., 2014a*;

Corresponding author
Matthew M. Rhodes,
mmrhodes@ualberta.ca

*Evangelista et al., 2014b*; *Palmer, 2014*; *Xu et al., 2015*; *Heers et al., 2016*; *Dececchi, Larsson & Habib, 2016*; *Sullivan, Xu & O'Connor, 2017*; *Segre & Banet, 2018*; *Talori et al., 2018*; *Talori et al., 2019*; *Pei et al., 2020*), and ventilatory structures (*Carrier & Farmer, 2000*; *Codd et al., 2008*; *Macaluso & Tschopp, 2018*), among other features. Without the ability to directly observe locomotion in extinct theropods, it is typically examined via osteological features, skeletal proportions, and trackway sites (*Gatesy, 1991*; *Gatesy & Middleton, 1997*; *Carrano, 1998*; *Carrano, 2000*; *Paul, 1998*; *Gatesy et al., 1999*; *Day et al., 2002*; *Milner, Lockley & Kirkland, 2006*). These methods are useful for inferring certain conditions but each has drawbacks as well. For example, although trackways directly record the motions of individual organisms, they may not represent characteristic behaviours or functional traits (e.g., speed, stride length, maneuverability). However, accounting for soft tissues that anchor and control the skeleton provides an additional viewpoint on locomotory adaptations.

Foundational studies of soft tissue reconstruction in dinosaurs were conducted in the early twentieth century, which used comparative methods grossly similar to extant phylogenetic bracketing but without categorical confidence levels that constrain speculation and improve consistency (*Romer, 1923a*; *Romer, 1923b*; *Romer, 1923c*; *Romer, 1927*; *Russell, 1935*). Comparisons with close living relatives provided data on muscles and their osteological landmarks that were then used to infer attachment sites on scarred surfaces, tubercles, and processes of fossil bones. Substantial research on patterns in locomotory evolution has been conducted since those foundational studies to reconstruct the musculature for various clades: abelisaurids (*Persons & Currie, 2011b*), allosauroids (*Bates & Schachner, 2011*; *Bates, Benson & Falkingham, 2012*), dromaeosaurids (*Perle, 1985*; *Hutchinson et al., 2008*; *Persons & Currie, 2012*), herrerasaurids (*Grillo & Azevedo, 2011*), ornithomimids (*Russell, 1972*), oviraptorosaurs (*Persons, Currie & Norell, 2014*), troodontids (*Bishop et al., 2018a*), tyrannosaurids (*Walker, 1977*; *Tarsitano, 1983*; *Perle, 1985*; *Carrano & Hutchinson, 2002*; *Hutchinson et al., 2005*; *Persons & Currie, 2011a*), and broadly among theropods (*Gatesy, 1990*; *Gatesy, 1995*; *Gatesy & Dial, 1996*; *Carrano & Biewener, 1999*; *Carrano, 2000*; *Farlow et al., 2000*; *Jones et al., 2000*; *Hutchinson & Gatesy, 2000*; *Persons & Currie, 2017*). Although reconstructing muscles in extinct taxa poses challenges (*McGowan, 1979*; *Bryant & Seymour, 1990*), proper phylogenetic context and categorization of confidence levels provide a stronger, more explicit framework for reconstruction (*Bryant & Russell, 1992*; *Witmer, 1995*). *Hutchinson (2001a)* and *Hutchinson (2001b)* used these more rigorous methods to produce a pair of thorough studies with a homology-based approach that documented osteological correlates of pelvic and femoral soft tissues in archosauromorphs. These two studies—plus follow-up summaries of hind limb muscle evolution in the same group (*Hutchinson, 2002*; *Hutchinson & Allen, 2009*)—all documented a stepwise accumulation of avian-like traits on the evolutionary line to birds. They have become oft-cited references for locomotory muscle reconstruction in dinosaurs, including musculoskeletal models that go beyond description of muscle scars to provide another dimension on locomotion studies (*Langer, 2003*; *Maidment & Barrett, 2011*; *Maidment & Barrett, 2012*; *Bates et al., 2012*; *Maidment, Bates & Barrett, 2014*; *Maidment et al., 2014*; *Bishop et al., 2018a*).

In theropods, many studies seem to corroborate the hypothesis that they represent a gradual continuum of morphological change regarding hind limb anatomy and locomotor function (*Hutchinson & Allen, 2009*). However, some have noted that anatomical changes (e.g., whole-body proportions, centre of mass position, size and scaling of appendicular skeleton relative to body mass) underwent significant evolutionary rate shifts in morphological evolution within Maniraptora that continued into early birds (*Allen et al., 2013*; *Dececchi & Larsson, 2013*; *Brusatte et al., 2014*), coincident with the development of feathers, opisthopuby, and other features (*Foth, Tischlinger & Rauhut, 2014*; *Xu et al., 2014*; *Sullivan, Xu & O'Connor, 2017*; *Macaluso & Tschopp, 2018*). To further complicate matters, locomotor abilities inferred from skeletal proportions and foot morphology indicate that convergence is common, such as the acquisition of relatively long distal limb elements (*Gatesy & Middleton, 1997*; *Carrano, 1998*; *Carrano, 1999*; *Persons & Currie, 2016*) or proximally compressed metatarsals (*Holtz, 1994*; *Snively & Russell, 2001*; *Snively & Russell, 2003*; *Snively, Russell & Powell, 2004*; *White, 2009*). Nevertheless, soft tissue reconstruction provides another perspective on these issues. In eudromaeosaurians, reduced extensor musculature corroborates osteological inferences suggesting reduced pursuit ability (*Carrano, 1999*; *Persons & Currie, 2012*; *Persons & Currie, 2016*). Conversely, tyrannosaurids exhibit several skeletal features associated with enhanced running and/or agility (*Holtz, 1994*; *Paul, 1998*; *Snively & Russell, 2001*; *Snively & Russell, 2003*; *Henderson & Snively, 2004*; *Persons & Currie, 2016*), but musculoskeletal models accounting for soft tissues suggest that running was improbable in large adults (*Hutchinson & Garcia, 2002*; *Hutchinson, 2004b*; *Hutchinson et al., 2005*; *Sellers et al., 2017*; *Dececchi et al., 2020*). Despite these advances, the locomotory musculature of many theropods is unknown, and recent discoveries of new theropods—particularly within Maniraptora—may require revisions of evolutionary patterns in light of previously unidentified conditions.

Given that soft tissue inferences may corroborate or conflict with osteological ones, how well does the myology of maniraptorans conform to the purported stepwise accumulation of avian-like features? To address this question, maniraptoran pelvic anatomy was inspected for osteological correlates of soft tissues, focusing on caenagnathids, therizinosaurs, and troodontids as taxa with understudied or unstudied pelvic myology (Fig. 1). Subsequently, the area of each muscle origin may be quantified and compared, with functional inferences inferred from previous studies or the most closely related taxa. Examination of the hip focuses on proximal hind limb muscles with a primary relationship to locomotion that protract and retract the entire leg, and flex and extend joints of the leg (*Dilkes, 2000*; *Meers, 2003*; *Maidment & Barrett, 2011*). Furthermore, the pelvis of a biped is the junction between the axial body and the locomotory system. In particular, the pelvis sits at the junction of the hind limb and tail, and provides key information on caudal locomotory musculature and its degree of integration with the thigh (*Gatesy, 1990*; *Gatesy & Dial, 1996*; *Persons & Currie, 2011a*; *Persons & Currie, 2011b*).

However, there is evidence that the morphology of an attachment site (an enthesis) does not correspond muscular excursion, force production, or other attributes of its associated muscle (*Zumwalt, 2006*; *Rabey et al., 2015*). Because of this, the area of attachment data here may be unsuitable for reconstruction of other muscle properties including but not

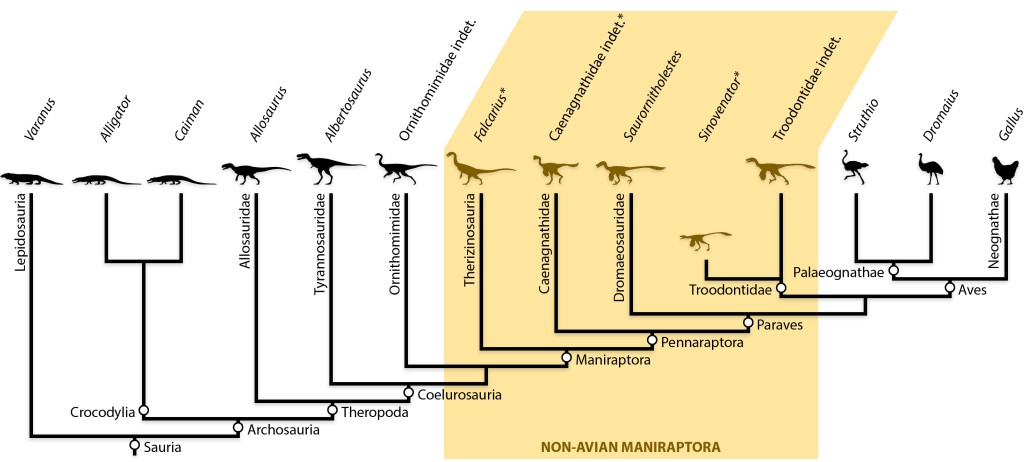

**Figure 1 Simplified phylogeny of non-avian maniraptorans among other theropods and extant relatives.** Taxa along top indicate what each reconstruction represents, and asterisks (*) indicate novel pelvic muscle reconstructions. See Methods for details on phylogenetic treatment of study taxa. Phylogeny based on *Hendrickx, Hartman & Mateus (2015)*.

limited to muscle volume, physiological cross-section area, force production, or activity level. Instead, the area of attachment for homologous muscles is herein presented as an opportunity to examine relative changes in a comparative context and investigate potential relationships with other aspects of palaeobiology. Shifts in the relative areas of each muscle or muscle group should not be used to directly infer muscle volume, force production, or other physiological attributes.

## MATERIALS & METHODS

Pelvic material of non-avian maniraptorans was examined, along with comparisons to that of other theropods, and to that of extant avians, crocodylians, and squamates (Table 1). In addition, a savannah monitor (*Varanus exanthematicus*), a spectacled caiman (*Caiman crocodilus*), and a common raven (*Corvus corax*) were dissected and inspected for pelvic and hind limb soft tissues (all UAMZ). Literature review provided a guide for dissection of pelvic musculature in squamates (*Russell & Bauer, 2008*; *Dick & Clemente, 2016*), crocodylians (*Romer, 1923a*; *Otero, Gallina & Herrera, 2010*; *Allen et al., 2015*), and birds (*Shufeldt, 1890*; *Hudson, Lanzillotti & Edwards, 1959*; *Halvorson, 1972*; *Ghetie, 1976*; *Jacobson & Hollyday, 1982*; *Baumel, Wilson & Bergren, 1990*; *Baumel et al., 1993*; *Mellett, 1994*; *Patak & Baldwin, 1998*; *Verstappen, Aerts & De Vree, 1998*; *Gangl et al., 2004*; *Smith et al., 2006*; *Smith et al., 2007*; *Paxton et al., 2010*; *Lamas, Main & Hutchinson, 2014*; *Hutchinson et al., 2015*). The homologies of pelvic musculature are summarized in Table 2. Material was photographed with a digital camera and illustrations were created in Adobe Photoshop or Illustrator CS6 using a Wacom tablet.

A guiding principle of the present study is the Extant Phylogenetic Bracket (*Bryant & Russell, 1992*; *Witmer, 1995*). This approach utilizes the closest extant representatives of an extinct focal taxon as the best evidence for soft tissue inferences. Dissections

**Table 1  Pelvic material examined for osteological correlates of soft tissues.**

| Taxon | Specimen number(s) | Pelvic element(s) |
|---|---|---|
| Non-maniraptoran Theropoda | | |
| *Albertosaurus sarcophagus* | CMN 11315 | Ilia, fused pubes, ischium |
| *Allosaurus fragilis* | BYUVP 4891, 5111, 5292, 11430, 13625, 16774, 17550; CMN 38454 | Ilia, pubes, ischia |
| *Ceratosaurus* sp. | BYUVP 12893 | Fused pubes, ischium |
| *Daspletosaurus torosus* | UALVP 52981 | Articulated pelvis |
| *Dromiceiomimus brevitertius* | ROM 797; UALVP 16182 | Fused pubes (ROM); Semi-articulated pelvis (UALVP) |
| *Gorgosaurus libratus* | UALVP 10 | Articulated pelvis (fused pubes broken midshaft) |
| *Ornithomimus edmontonicus* | ROM 851 | Fused pubes |
| *Torvosaurus tanneri* | BYUVP 2013, 2014, 2015, 4881, 4977 | Ilia, pubes, ischia |
| Ornithomimidae indet. | CMN 8897, 12348; TMP 1967.020.0230, 1967.020.0237, 1981.016.0679, 1981.022.0025, 1989.036.0103, 1992.050.0065, 1994.012.0428, 1996.012.0019, 2009.035.0001, 2013.012.0007; UALVP 50646, 60331 | Ilium, partial ischia (CMN); ilium, pubes, ischia, articulated pelvis (TMP); partial ilium, complete ischium (UALVP) |
| Therizinosauria | | |
| *Falcarius utahensis* | UMNH VP 12370, 12371, 12374, 14540, 14659, 31071 | Pubes, ischia |
| *Falcarius* sp. | CEUM 52424, 52482, 52520, 53243, 53252, 53305, 53312, 53349, 53361, 73681, 73709, 73963, 73964, 74706, 74717, 74727, 74739, 74763, 74794, 74842, 77035, 77037, 77045, 77051, 77053, 77081, 77114, 77173, 77189, 77194, 77195, 77233, 77241, 77290, 78223 | Ilia, pubes, ischia |
| Caenagnathidae | | |
| *Anzu wyliei* | CM 78000 | Partial ilium, pubes, ischia |
| *Chirostenotes pergracilis* | TMP 1979.020.0001, TMP 2002.012.0103 | Ilia, ischium |
| *Epichirostenotes curriei* | ROM 43250 | Ilia, right ischium, pubes |
| Caenagnathidae indet. | TMP 1980.016.2095, 1981.023.0034–35, 1982.016.0275, 1992.036.0674, 1994.012.0603, 1998.093.0013; UALVP 56638, 59791 | Ilia, disarticulated and fused pubes (TMP); partial ilium, fused pubes (UALVP) |
| Dromaeosauridae | | |
| *Hesperonychus elizabethae* | UALVP 48778 | Partial articulated pelvis |
| *Saurornitholestes langstoni* | MOR 660; UALVP 55700 | Ilium (MOR); articulated pelvis (UALVP) |
| *Utahraptor ostrommaysorum* | BYUVP 14302, 19973, 20692 | Partial ilia, pubis |
| Dromaeosauridae indet. | TMP 1986.077.0002 | Ischium |
| Velociraptorinae indet. | ROM 53573 | Cast of articulated pelvis |
| Troodontidae | | |
| *Jianianhualong tengi* | DLXH 1218 | Semi-articulated pelvis |
| *Latenivenatrix mcmasterae* | UALVP 55804 | Ilia, partial pubes |
| *Saurornithoides mongoliensis* | AMNH FR 6516 | Ischia |
| *Sinovenator changii* | IVPP V12583, V12615 | Ilia, pubes |
| *Talos sampsoni* | UMNH VP 19479 | Partial ilium, partial pubes, partial ischia |

**Table 1** (*continued*)

| Taxon | Specimen number(s) | Pelvic element(s) |
|---|---|---|
| Aves | | |
| *Apteryx haastii* | RM 8369 | Articulated pelvis |
| *Casuarius australis* | UAMZ 1369 | Articulated pelvis |
| *Dromaius novaehollandiae* | ROM R6843, R7654; UAMZ B-FIC2014.260 | Articulated pelves |
| *Gallus gallus* | RM 8355 | Articulated pelvis |
| *Rhea americana* | RM 8499 | Disarticulated pelvis |
| *Struthio camelus* | ROM R1080, R1162, R1933, R2136, R2305; UAMZ 7159 | Disarticulated and articulated pelves |
| Crocodylia | | |
| *Alligator mississippiensis* | ROM R343 | Semi-articulated pelvis |
| *Alligator* sp. | UAMZ HER-R654 | Articulated pelvis |
| *Caiman crocodilus* | RM 5242 | Disarticulated pelvis |
| *Osteolaemus tetraspis* | RM 5216 | Articulated pelvis |
| Squamata | | |
| *Tupinambis teguixin* | ROM R436 | Articulated pelvis |
| *Varanus albigularis* | RM 5220 | Disarticulated pelvis |
| *Varanus jobiensis* | RM 5219 | Articulated pelvis |
| *Varanus komodoensis* | ROM R7565 | Disarticulated pelvis |
| *Varanus niloticus* | RM 5221 | Articulated pelvis |
| *Varanus rudicollis* | ROM R7318 | Articulated pelvis |
| *Varanus salvator* | RM 5222, 5223, 5224 | Articulated pelves |

and comparative anatomy of skeletonized specimens provided data on soft tissues and their osteological correlates in living relatives of non-avian maniraptorans. Following previous studies (*Hutchinson, 2001a*; *Carrano & Hutchinson, 2002*; *Hutchinson et al., 2005*; *Hutchinson et al., 2008*; *Maidment, Bates & Barrett, 2014*; *Bishop et al., 2018a*), these data were used to examine pelvic material of non-avian maniraptorans for osteological correlates of soft tissues, which were then mapped onto a reconstructed pelvis in each taxon. For taxa in which an articulated pelvis was unavailable or non-existent, the reconstruction was derived from associated material where possible, or from contact surfaces (i.e., joints between peduncles of pelvic bones) using well-preserved, closely related taxa for reference. It is unclear which subspecies is represented by the specimen of *Gallus* used in this study (Table 1); specimens influenced by artificial selection or commercialization may affect how accurately they represent galliform morphology compared to ancestral *Gallus* stock, which is perhaps better embodied by junglefowl (*Paxton et al., 2010*; *Paxton et al., 2014*). Previous studies note variation in body mass, muscle mass and architecture, locomotor ability, and force-generating capacity between junglefowl and commercial chickens (*Paxton et al., 2010*; *Paxton et al., 2014*; *Rose, Nudds & Codd, 2016*). However, these breeds tend to exhibit comparable limb muscle masses when each is expressed as a proportion of body mass (*Paxton et al., 2010*; *Rose, Nudds & Codd, 2016*). Moreover, it is not clear how these behavioural and physiological differences are related to enthesis morphology, if at all (*Zumwalt, 2006*; *Rabey et al., 2015*). Therefore, these potential

**Table 2  Homologies and abbreviations of pelvic musculature by anatomical muscle group (adapted from *Hutchinson, 2001a*).**

| Muscle | Lepidosauria | Crocodylia | Non-avian Theropoda | Aves |
|---|---|---|---|---|
| Triceps femoris | | | | |
| M. iliotibialis (IT)/Mm. iliotibiales cranialis et lateralis (IC+IL) | IT | IT1 | IT1 | IC |
| | [a] | IT2 | IT2 | IL |
| | [a] | IT3 | IT3 | [a] |
| M. ambiens (AMB) | AMB | AMB1+2 | AMB | AMB1+2 |
| M. iliofibularis | ILFB | ILFB | ILFB | ILFB |
| Deep dorsal group | | | | |
| M. iliofemoralis (IF)/Mm. iliotrochantericus caudalis et iliofemoralis externus (ITC+IFE) | IF | IF | ITC | ITC |
| | [a] | [a] | IFE | IFE |
| M. puboischiofemoralis internus (PIFI)/Mm. iliofemoralis internus (IFI) and iliotrochanterici cranialis et medius (ITCR+ITM) | PIFI1+2 | PIFI1 | PIFI1 | IFI |
| | PIFI3 | PIFI2 | PIFI2 | ITCR |
| | [a] | [a] | [a] | ITM |
| Flexor cruris group | | | | |
| M. puboischiotibialis (PIT) | PIT1 | – | – | – |
| | PIT2[b] | PIT | – | – |
| | PIT3[b] | FTI2 | – | – |
| M. flexor tibialis internus (FTI)/M. flexor cruris medialis (FCM) | FTI1 | FTI1 | FTI1 | – |
| | FTI2 | FTI3 | FTI3 | FCM |
| | [a] | FTI4[b] | – | – |
| M. flexor tibialis externus (FTE)/M. flexor cruris lateralis pars pelvica (FCLP) | FTE[b] | FTE | FTE | FCLP |
| M. pubotibialis (PUT) | PUT | – | – | – |
| M. adductor femoris (ADD)/Mm. puboischiofemorales medialis et lateralis (PIFM+PIFL) | ADD | ADD1 | ADD1 | PIFM |
| | [a] | ADD2 | ADD2 | PIFL |
| M. puboischiofemoralis externus (PIFE)/Mm. obturatorii lateralis et medialis (OL+OM) | PIFE | PIFE1 | PIFE1 | OL |
| | [a] | PIFE2 | PIFE2 | OM |
| | [a] | PIFE3 | PIFE3 | – |
| M. ischiotrochantericus (ISTR)/M. ischiofemoralis (ISF) | ISTR | ISTR | ISTR | ISF |
| M. caudofemoralis brevis (CFB)/M. caudofemoralis pars pelvica (CFP) | CFB | CFB | CFB | CFP |

**Notes.**
⁻absent.
[a]undivided.
[b]origin located on soft tissue, not directly on pelvis.

differences between specimens of *Gallus* are not expected to substantially influence our myological reconstruction or quantitative comparisons that account for body mass.

For Therizinosauria, published descriptions of various taxa were reviewed (*Xu, Tang & Wang, 1999*; *Kirkland et al., 2005*; *Li, You & Zhang, 2008*; *Zanno et al., 2009*; *Zanno, 2010a*; *Zanno, 2010b*; *Pu et al., 2013*; *Yao et al., 2019*), but only specimens pertaining to *Falcarius* were used in muscle reconstruction. There exists the potential that some of these specimens are taxonomically different at the specific or even generic level (*Zanno, 2010a*), but limited variation was observed in the osteological correlates of the specimens examined; the taxonomic differences therefore are not expected to significantly influence the results.

Caenagnathids exhibit some variation, chiefly among the ilia (*Funston & Currie, 2020*; *Rhodes, Funston & Currie, 2020*). The relatively poor preservation of their pelvic elements required comparison to previous descriptions for reconstruction, including those of closely related oviraptorosaurs (*Currie & Russell, 1988*; *Sues, 1997*; *Barsbold et al., 2000*; *Clark, Norell & Barsbold, 2001*; *Lü, 2002*; *Lü et al., 2004*; *Lü et al., 2013*; *Lü et al., 2017*; *Sullivan, Jasinski & Tomme, 2011*; *Xu et al., 2013*; *Lamanna et al., 2014*; *Funston et al., 2018*). The reconstructed pelvis and musculature presented here relies heavily on material of *Chirostenotes* (*Currie & Russell, 1988*; *Funston & Currie, 2020*) but serves as a general blueprint for Caenagnathidae given the minor pelvic variation within the family.

In troodontids, variation in pelvic morphology warranted two separate reconstructions based on observed fossils. More basal, Early Cretaceous troodontids, such as *Sinovenator* (*Xu et al., 2002*), exhibit similar morphology and osteological correlates with each other (*Russell & Dong, 1993*; *Currie & Dong, 2001*; *Xu et al., 2017*; *Pei et al., 2020*), but differ substantially from those of Late Cretaceous forms (*Norell et al., 2009*; *Zanno et al., 2011*; *Tsuihiji et al., 2014*; *Van der Reest & Currie, 2017*; *Pei et al., 2017*). Basal and derived troodontids were reconstructed and are herein treated separately based on these differences. However, basal members are paraphyletic with respect to derived taxa (*Xu et al., 2002*; *Xu et al., 2011*; *Xu et al., 2017*; *Zanno et al., 2011*; *Tsuihiji et al., 2014*; *Hendrickx, Hartman & Mateus, 2015*; *Shen et al., 2017b*) and thus useful to polarize anatomical changes within the family. They are intended to represent subsequent stages—not equivalent taxonomic groups—within Troodontidae (Fig. 1).

Although the areas of pelvic muscle origins may not be an ideal or even suitable method to determine other properties of muscles (*Zumwalt, 2006*; *Rabey et al., 2015*), it does allow for examination of relative changes that occurred across non-avian theropods. Other methods, such as analysis of cancellous bone architecture (*Bishop et al., 2018a*), can produce dynamic models and informative results, but are limiting because sampling is not possible for many specimens. Comparison of origin areas can provide data across a wide range of taxa, but this only represents one aspect of locomotion. Accordingly, these data are limited and cannot unequivocally determine locomotory adaptations or directly inform on other attributes of muscle function. Nevertheless, these data can be incorporated into a more holistic analysis of locomotion and behaviour in theropods. The areas of pelvic muscle origins (Figs. S1–S14) were calculated using two methods. In the first method, the image of a pelvis with its associated muscle regions was imported into the illustration program Corel DRAW!, and the perimeters of each origin were traced counter-clockwise using the polyline tool to generate a contour with a positive (right hand) sense of curl. Each contour was then exported as an AutoCAD DXF file and the area enclosed by the contour was then

calculated with some custom software. More details of this method can be seen with the determination of the areas of theropod orbits (*Henderson, 2002*). In the second method, diagrams of pelvic muscles were imported to ImageJ, the scale set, and then each area of origin measured using the wand (tracing) tool under default settings. In both methods, a muscle origin that appeared in multiple views had its portions summed. This relies on the assumption that using standard anatomical views, which present each view rotated 90° from one another, adequately captures the surface area covered by the origin if flattened to a 2-dimensional shape. In addition, the pubes in a non-avian theropod are fused together, so the anterior and posterior views of these bones display both the anatomical left and right sides of the pelvis (Figs. S4–S11). This was mimicked in the figures of the crocodylians for consistency and more anatomically realistic diagrams. However, only the origins appearing on one anatomical side of the pelvis were measured. This is because pelvic musculature only affects the hind limb on the same side of the body and is identical on the contralateral side of a bilaterally symmetric organism.

R statistical software (*R Core Team, 2020*) was used to test whether the muscle area results from the Corel DRAW! tracings and ImageJ measurements were significantly different. The absolute areas of pelvic muscle origins measured by each method (Table S1) were compared using a two-sample $t$-test for each taxon individually and all taxa collectively. To measure the similarity between results from Corel DRAW! and ImageJ, these matrices (Table S2–S3) were subjected to a RV coefficients analysis (*Robert & Escoufier, 1976*) performed in R using the 'MatrixCorrelation' package v. 0.9.2 (*Liland, 2017*; *Indahl, Næs & Liland, 2018*). A sensitivity analysis was conducted by performing a one-way ANOVA on measurements of *Albertosaurus* gathered in ImageJ under a variety of tolerance levels (thresholds; 0 [default], 16, 32, 48, 64) to examine the effect of tolerance level on measured origin areas (Table S4). *Albertosaurus* was chosen because its total area of pelvic origins was greatest in absolute value and included origins of small and large areas, and thus was expected to show the highest variation under different tolerance levels.

Functional groups were compared in antagonistic pairs (flexor/extensor, abductor/adductor, medial/lateral rotator) by the proportion of combined origin areas in each functional group (e.g., flexors) to the total area of both antagonistic functions (e.g., flexors + extensors). Muscles were assigned to functional groups based on previous studies of locomotion and their function(s) were averaged. The inferred functions were determined in a "consensus" framework in which the functions of each muscle were compared and amalgamated across these studies. In each of the extant groups, one study or review was used as the primary source as it examined functions in all three spatial dimensions by directly measuring activity or by using detailed musculoskeletal models to measure moment arms. This was given preference over studies that hypothesized or estimated actions based on gross anatomy, which provided complementary data but tended to assess fewer spatial dimensions and comprised subsets of the functions presented in the primary source in nearly all cases. Conflicts were rare, but if they arose, preference was given to studies that directly measured function, then to moment arm analyses, and then to estimation from gross anatomy. Within this framework, locomotion in squamates primarily followed *Russell & Bauer (2008)* alongside other sources (*Rewcastle, 1983*; *Higham & Jayne,*

*2004*; *Dick & Clemente, 2016*). In crocodylians, functions were ascertained from *Bates & Schachner (2011)* and supplemented by others (*Gatesy, 1997*; *Allen et al., 2015*). Among non-avian theropods, data were available for *Allosaurus* (*Bates, Benson & Falkingham, 2012*), *Tyrannosaurus* (*Hutchinson et al., 2005*), *Velociraptor* (*Hutchinson et al., 2008*), and all of these taxa plus *Struthiomimus* (*Bates & Schachner, 2011*). The functions for *Struthio* relied mainly on *Hutchinson et al. (2015)* with consideration of *Smith et al. (2006)*, although conflict was noted for the actions of the Mm. obturatorii lateralis et medialis and discussed in the former of these two texts. One detailed study provided data for *Dromaius* (*Lamas, Main & Hutchinson, 2014*), and one main paper was used for *Gallus* (*Paxton et al., 2010*) alongside a pair of complementary papers (*Gatesy, 1999a*; *Gatesy, 1999b*). For any extinct taxon lacking data, functional roles were assigned based on comparison to bracketing taxa with data.

To analyze the distributions of cursorial categories, Jenks Natural Breaks optimization in the XRealStats package for Microsoft Excel was used following *Powers, Sullivan & Currie (2020)*. The distributions were analyzed across four classes ($k = 4$) to replicate traditional cursorial categories sensu *Carrano (1999)*, and across five classes ($k = 5$) to test if increasing the number of categories better reflected the data in question. All tests had 1000 iterations regardless of the number of classes. Jenks Natural Breaks optimization calculates a goodness of variance fit (GVF) value in which a result closer to 1 indicates a better fit. The four- and five-class tests from Jenks Natural Breaks optimization were treated as models and each group was compared using the Akaike Information Criterion (AIC; *Akaike, 1973*), which evaluates the relative quality of each model. Because of the sample size of our data, the small-sample corrected version of this analysis was used (AIC$_C$; *Hurvich & Tsai, 1989*). The difference between the score of each model and the minimum score of all models ($\Delta$AIC$_C$) allows interpretation of relative model quality wherein lower scores indicate better fit. Models with $\Delta$AIC$_C \leq 2$ have substantial support, whereas models with $\Delta$AIC$_C$ >10 have practically no support (*Burnham & Anderson, 2002*; *Burnham & Anderson, 2004*). The potential effects of allometry on the results were explored by comparing body size to the total area of all hip musculature, the area of major extensors only, and the length of the ilium (as a proxy for pelvis size). Body mass estimates were calculated from stylopod circumferences using equation 1 of *Campione & Evans (2012)* for quadrupeds and equation 7 of *Campione et al. (2014)* for bipeds. Wherever possible, the same specimen used for the reconstruction of pelvic musculature was used to estimate body size. In cases where this was not possible, a specimen of similar size was used. For the pair of crocodylians, stylopod circumferences were not measured, but could be estimated based on the equations provided by *Dodson (1975)* to calculate minimum diameter. Data were already available for some non-avian theropods including *Allosaurus* (*Bates et al., 2009a*), *Dromiceiomimus* (*Campione & Evans, 2020*), *Falcarius* (*Campione & Evans, 2020*), *Chirostenotes* (*Funston, 2020*), *Saurornitholestes* (*Campione & Evans, 2020*), and a derived troodontid (*Benson et al., 2014*). Body mass estimates using the bipedal formula were significantly higher than the range of each avian species (*Dunning Jr, 2007*), so these estimates were replaced with species averages. Because of differences in mean body mass depending on sex, either the male or female mean was selected from published sources (*Dunning Jr, 2007*; *Olson &*

*Turvey, 2013*) based on the primary specimen used in myological reconstruction of each palaeognath—an adult male *Struthio* (UAMZ 7159) and adult female *Dromaius* (UAMZ B-FIC2014.260). For the chicken, the body mass of the adult *Gallus* specimen in *Allen et al. (2013)* was used because it closely matched adults of other studies (*Paxton et al., 2010*; *Paxton et al., 2014*). Body mass estimates are provided with a ±25% range, which seems to generally correspond to the ranges of the extant avian species in this study. All variables were log-transformed to normalize the distribution of the dependent variable (*Currie, 2003*) and then subjected to a Phylogenetic Generalized Least Squares (PGLS) regression analysis using the 'caper' package v. 1.0.1 in R. In all of the bivariate comparisons, the dependent variable was set against estimated body mass to standardize the data to body size (Table S5). Supplemental files include the R script associated with all calculations and a nexus file containing the tree for PGLS regressions that matches the topology of Fig. 1. All branches were equally weighted; a time-calibrated phylogeny is beyond the scope of this study and may not be very informative given our small sample size and potential skew caused by the shared geologic age of most extinct taxa examined.

## RESULTS

The inferred pelvic myology of the therizinosaur, caenagnathid, and troodontid specimens observed here is largely consistent with that of other theropods (*Hutchinson, 2001a*; *Carrano & Hutchinson, 2002*; *Hutchinson et al., 2005*; *Hutchinson et al., 2008*; *Bishop et al., 2018a*). Generally, origins of musculature track morphological changes in the bones, and the number and arrangement of pelvic muscles remain conservative throughout non-avian Maniraptora. However, a few osteological correlates in each of these clades differ—in some cases, notably—from existing literature and the evolutionary continuum on the line to birds (*Hutchinson, 2001a*; *Hutchinson & Allen, 2009*). Novel maniraptoran osteological and myological data are arranged primarily by increasing levels of inference (*Witmer, 1995*) from osteology to myology to inferred function, and then secondarily by phylogenetic order from stemward to crownward.

### Osteological correlates

*Falcarius.* The therizinosaur *Falcarius* has ilia characterized as altiliac (*Zanno, 2010a*), not dolichoiliac as in more basal theropod clades such as tyrannosaurids or ornithomimids (Fig. 2A). The cuppedicus (= preacetabular) 'fossa' lacks a bony medial wall (*Zanno, 2010a*), but the medial side of the preacetabular hook has a concavity delineated dorsally by a low, arcuate ridge confluent with a narrow, medially projecting shelf (Figs. 2C–2D). This concavity is covered in striae that radiate anteriorly from the shelf in ventrally-concave arcs toward the anterior margin of the ilium and the tip of the preacetabular hook (Fig. 2C). The brevis fossa is narrow and shallow, which strongly contrasts with the wide and deep brevis fossae of more plesiomorphic theropods (*Osborn, 1903*; *Hutchinson, 2001a*; *Carpenter et al., 2005*). Osteological correlates of the pubis are consistent with other theropods (*Hutchinson, 2001a*). The preacetabular tubercle is reduced but an oval patch of rugose texture is adjacent and lateral to it (Figs. 2A, 2E), and the pubic apron has longitudinal striations covering its anterior and posterior sides (Figs. 2G–2I). The ischium is reduced

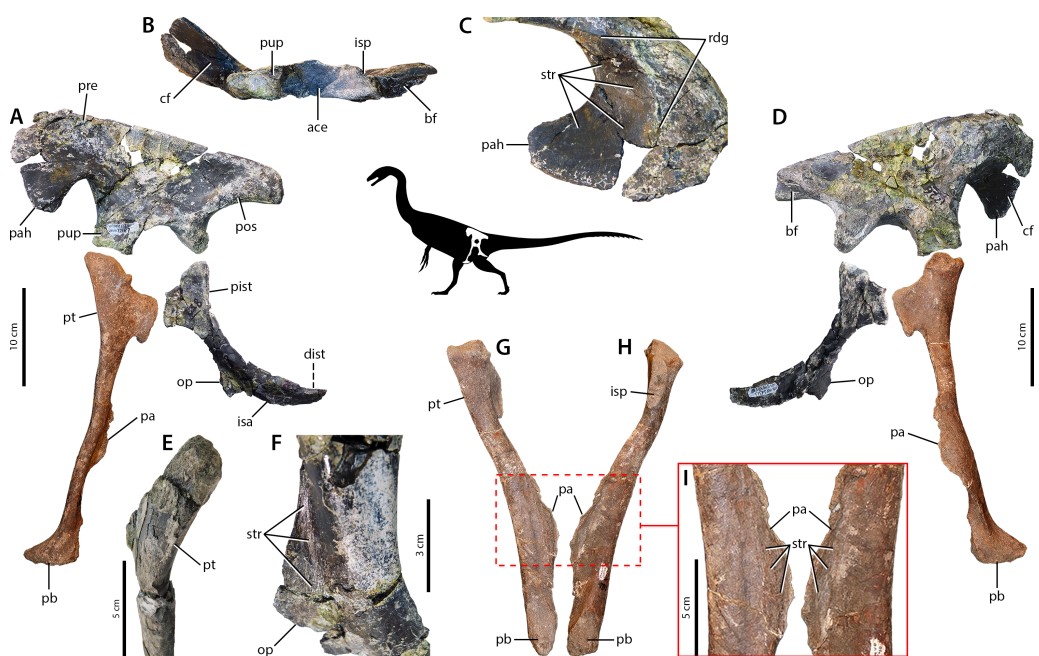

**Figure 2** **Osteological correlates of pelvic musculature in *Falcarius*.** Pelvis in left lateral view (A) with ilium CEUM 77189, pubis UMNH VP 14540 (reversed; courtesy of Natural History Museum of Utah), and ischium CEUM 74717 (reversed). Ilium CEUM 77189 in ventral view (B) and oblique view of cuppedicus fossa (C). Pelvis in medial view (D) with the same specimens as in lateral view. Anterolateral view of proximal end of pubis CEUM 52424 (E). Close-up of obturator process (F) of ischium CEUM 52482 (reversed; proximal to top). Pubis UMNH VP 14540 in anterior (G) and posterior (H) views with close-up of the apron (I). Abbreviations: ace, acetabulum; bf, brevis fossa; cf, cuppedicus fossa; dist, distal ischial tubercle; isa, ischial apron; isp, ischiadic peduncle; op, obturator process; pa, pubic apron; pah, preacetabular hook; pb, pubic boot; pist, proximal ischial tubercle; pos, postacetabulum; pre, preacetabulum; pt, preacetabular tubercle; pup, pubic peduncle; rdg, ridge; str, striations.

relative to non-maniraptoran theropods, but exhibits similarity in gross morphology to other maniraptorans (*Zanno, 2010a*). This includes reduced proximal and distal ischial tuberosities (Fig. 2A) and proximodistally oriented striations on the lateral surface of the obturator process (Fig. 2F).

Caenagnathidae indet. The osteological correlates of the pelvic musculature of caenagnathids are detailed in *Rhodes, Funston & Currie (2020)*, and the morphological variation exhibited within the family does not strongly affect the general layout and arrangement of pelvic muscle correlates. Whereas most of these correlates are congruent with other theropods (*Hutchinson, 2001a*; *Hutchinson, 2002*; *Hutchinson et al., 2005*; *Hutchinson et al., 2008*; *Bishop et al., 2018a*), the general anatomy and noteworthy correlates are emphasized here. Caenagnathids are dolichoiliac as the postacetabulum is reduced relative to the preacetabulum (Figs. 3A, 3D) (*Currie & Russell, 1988*; *Funston & Currie, 2020*; *Rhodes, Funston & Currie, 2020*). Much like *Falcarius*, the cuppedicus 'fossa' lacks a bony medial wall—only bordered dorsally by a narrow ridge that merges into the concave medial side of the preacetabular hook (Figs. 3C, 3F)—and the brevis fossa is narrow, shallow, and reduced compared to non-maniraptoran theropods (Figs. 3E–3F) (*Osborn,*

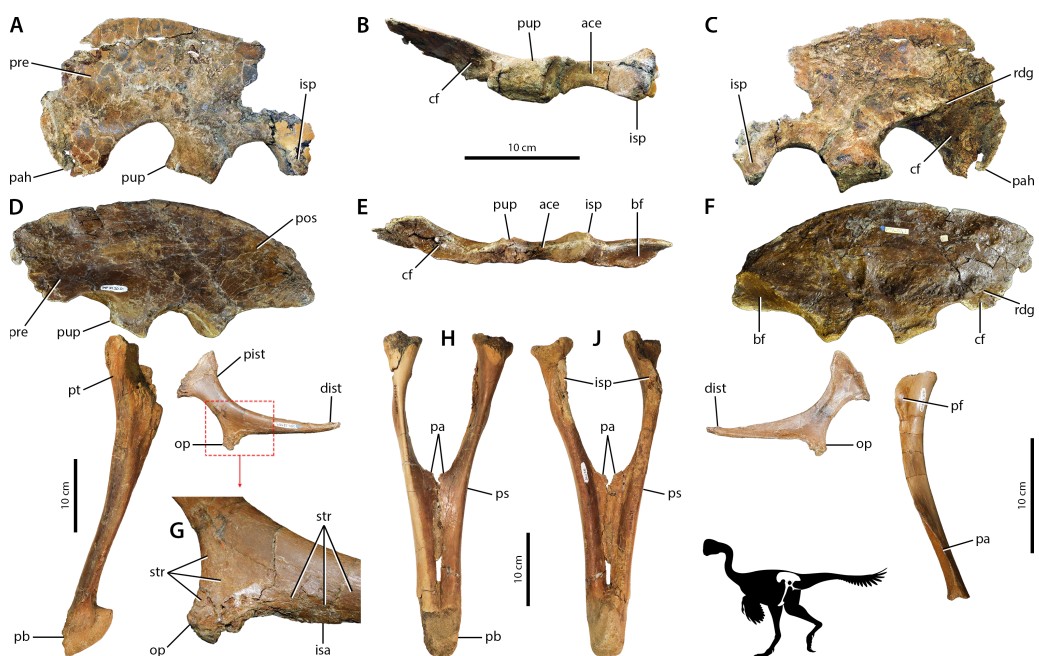

**Figure 3** **Osteological correlates of pelvic musculature in Caenagnathidae indet.** Ilium UALVP 59791 in left lateral (A), ventral (B), and medial (C) views. Pelvis in left lateral view (D) with ilium TMP 1979.020.0001, pubis UALVP 56638, and ischium TMP 1979.020.0001 (reversed). Ilium TMP 1979.020.0001 in ventral view (E). Pelvis in medial view (F) with the same specimens as in lateral view except for pubis TMP 1980.016.2095. Close-up of obturator process (G) of ischium TMP 1979.020.0001. Pubes UALVP 56638 in anterior (H) and posterior (J) views. Refer to Fig. 2 for anatomical abbreviations.

*1903*; *Gatesy, 1990*; *Carpenter et al., 2005*; *Persons & Currie, 2011a*). As identified in *Rhodes, Funston & Currie (2020)*, the anterior side of the pubic apron possesses longitudinal striations but the posterior surface lacks muscle scars, whereas longitudinal striations cover the posterior edge of the pubic shaft. The ischium is also comparable to *Falcarius* because it is reduced relative to non-maniraptoran theropods, as are the proximal and distal ischial tuberosities (Fig. 3D).

*Sinovenator*. Basal troodontid characteristics are expressed in *Sinovenator* (*Xu et al., 2002*). The pelvic morphology of this troodontid is readily comparable to other basal members such as *Mei* (*Xu & Norell, 2004*; *Gao et al., 2012*; *Pei et al., 2020*). The ilium is small with an anteroposteriorly short preacetabular blade, in which a small portion of the anterior edge is broken away (Figs. 4A–4B) that has been reconstructed conservatively based on other basal troodontids (*Russell & Dong, 1993*; *Currie & Dong, 2001*). A shallow, circular depression sits on the lateral side of the anteroposteriorly long, dorsoventrally deep pubic peduncle at the base of the preacetabular hook (Figs. 4A–4B). The rugose dorsal iliac margin continues into the postacetabular blade, which tapers posteriorly and has striae on its posterolateral corner (Figs. 4A–4B) (*Shen et al., 2017a*). The brevis fossa is somewhat reduced into a narrow, shallow concavity (Figs. 4C–4D). On the pubic apron, both the anterior and posterior sides are covered in longitudinal striae (Figs. 4G–4J). Posteriorly, the pubic apron forms a transversely broad trough or embayment that extends laterally so

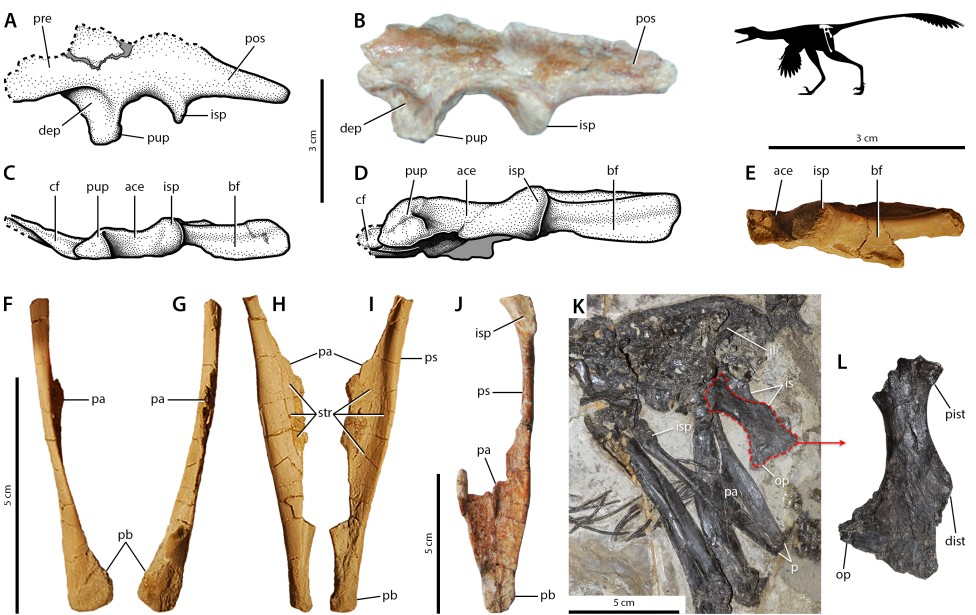

**Figure 4** Osteological correlates of pelvic musculature in *Sinovenator* (A–J) and *Jianianhualong* (K–L). Ilia IVPP V12615 and IVPP V12583 in left lateral (A–B) and ventral (C–E) views (A, C, and E reversed). In the stippled line drawings, dashed lines indicate broken edges and grey represents matrix-obscured areas. Pubis IVPP V12583 in lateral (F, reversed), medial (G, reversed), anterior (H), and posterior (I) views. Pubes IVPP V12615 in posterior view (J). Pelvic region of *Jianianhualong* (K) with close-up of left ischium DLXH 1218 (L). Refer to Fig. 2 for anatomical abbreviations.

that the posterior margins of the shafts are mediolaterally compressed into longitudinal ridges (Figs. 4H–4J). The ischium is small, has a distally positioned obturator process, and exhibits conspicuous processes along the posterior margin (Figs. 4J–4K) (*Russell & Dong, 1993*; *Currie & Dong, 2001*; *Xu et al., 2002*; *Xu et al., 2017*). The lateral surface of the obturator process also shows scars (Fig. 4K).

Derived Troodontidae indet. Derived troodontids resemble *Gobivenator* (*Tsuihiji et al., 2014*) in their pelvic anatomy. Striations are preserved along the anterior and dorsal margins of the ilium, extending onto a rugose ridge along the dorsolateral edge of the blade (Figs. 5A, 5C). The postacetabulum tapers slightly and forms a rounded or squared-off posterior edge (*Tsuihiji et al., 2014*) (Fig. 5A). A triangular patch of striations covers the posterolateral corner of the ilium (Fig. 5A). The brevis fossa is moderately expanded—dorsoventrally and transversely—relative to those of more basal troodontids (Figs. 5B–5C), but not to the same extent as non-maniraptoran coelurosaurs (*Osborn, 1903*; *Carpenter et al., 2005*; *Persons & Currie, 2011a*). A prominent preacetabular tubercle projects anterolaterally from the iliac peduncle of the pubis, its anteroventral margin mediolaterally compressed into a spine-like crest with striae oriented parallel to the margin (Figs. 5A, 5F). Longitudinal striations cover the anterior and posterior surfaces of the pubic apron, which engulf the posterior edges of the shafts as well (Figs. 5G–5M). On the lateral side of the pubis of UALVP 55804, this striated region extends distally to a relatively small tubercle that may be the osteological correlate for a pubogastralial ligament (Figs. 5G–5H, 5J–5K) (*Rhodes*

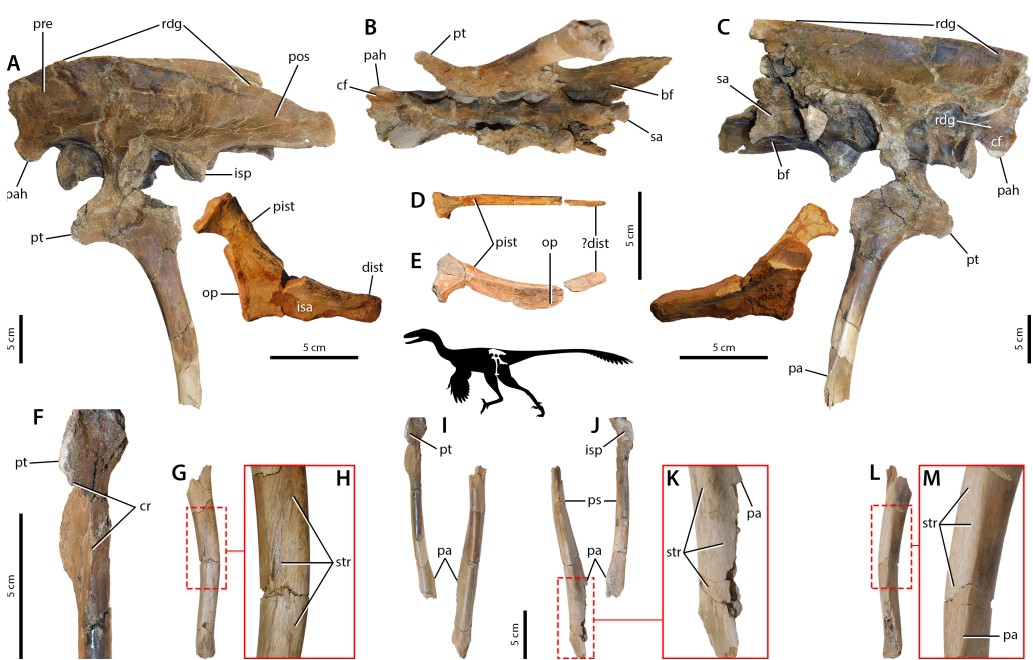

**Figure 5 Osteological correlates of pelvic musculature in derived Troodontidae indet.** Pelvis in left lateral view (A) with ilium and pubis UALVP 55804 (reversed) and ischium AMNH 6516. Pelvis UALVP 55804 (reversed) in ventral view (B). Pelvis in medial view (C) with the same specimens as in lateral view. Ischium UMNH VP 19479 (reversed; courtesy of Natural History Museum of Utah) in dorsal (D) and lateral (E) views. Pubis UALVP 55804 in anterior view (F). Pubes UALVP 55804 in lateral (G, close-up in H), anterior (I), posterior (J, close-up in K), and medial (L, close-up in M) views. Refer to Fig. 2 for anatomical abbreviations.

*& Currie, 2020*). The ischium is triangular (Fig. 5A) (*Norell et al., 2009*; *Zanno et al., 2011*). Whereas the obturator process is relatively larger than in more basal troodontids, the proximal and distal ischial tuberosities are reduced to low eminences (Figs. 5A, 5D–5E). All of these sites have a slightly rugose textures, and between the ischial tuberosities is another longitudinal region along the dorsolateral edge of the shaft (Figs. 5A, 5C–5E).

## Inferred myology

*Falcarius.* The origins of pelvic musculature in *Falcarius* (Fig. 6A) generally correspond to those of other theropods (Figs. 7D–7G) (*Hutchinson et al., 2005*; *Hutchinson et al., 2008*; *Bishop et al., 2018a*). On the ilium, the origin of M. puboischiofemoralis internus 1 on the anterolateral side of the pubic peduncle is clearly separate from that of M. puboischiofemoralis internus 2 (Fig. 6A), which is not the case in plesiomorphic relatives such as ornithomimids or tyrannosaurids (Figs. 7E–7F) (*Russell, 1972*; *Carrano & Hutchinson, 2002*; *Hutchinson et al., 2005*). Radiating striations on the medial side of the preacetabular hook indicate that the origin of the M. puboischiofemoralis internus 2 occupied this entire surface (Fig. 6A). The origin of M. caudofemoralis brevis is reduced relative to the same origin in more basal theropod lineages (Figs. 6A, 7D–7F) (*Carrano & Hutchinson, 2002*; *Hutchinson et al., 2005*; *Persons & Currie, 2011a*), reflecting similar reduction in the brevis fossa, its osteological correlate (Figs. 2B, 2D). On the pubis, the

origin of M. ambiens is not well defined, but nonetheless present on the oval patch of textured bone surface adjacent to the preacetabular tubercle (Fig. 6A). The origins of Mm. flexores tibiales interni 1 et 3 on the ischium are present but reduced, in turn with the reduction in the distal and proximal ischial tuberosities as their respective osteological correlates. Other origins of pelvic musculature similarly track morphological changes in the bones, but the layout of these origins are similar to other theropods (Figs. 6A, 7D–7G) (*Hutchinson et al., 2005*; *Hutchinson et al., 2008*; *Bishop et al., 2018a*).

Caenagnathidae indet. Caenagnathid pelvic musculature (Fig. 6B) shows many similarities to that reconstructed for *Falcarius*. The origin of M. puboischiofemoralis internus 2 likewise occupies the entire medial side of the preacetabular hook, and the origin of M. caudofemoralis brevis is also shrunken compared to the conditions in non-maniraptoran theropods (Figs. 6B, 7D–7F) (*Hutchinson et al., 2005*; *Persons & Currie, 2011a*). On the pubis, the origin of M. ambiens is also adjacent to the preacetabular tubercle, and the striated anterior surface of the pubic apron marks the origin of M. puboischiofemoralis externus 1 (Fig. 6B). However, the longitudinal striae on the posterior edges of the pubic shafts—instead of the pubic apron—demarcate the origin of M. puboischiofemoralis externus 2 (Fig. 6B), which exhibits a unique condition among Archosauria (*Rhodes, Funston & Currie, 2020*). Muscle attachment sites on the ischium and elsewhere on the pelvis do not vary considerably from those of *Falcarius* or other theropods (Figs. 6B, 7D–7G) (*Hutchinson, 2002*).

*Sinovenator.* The organization of most pelvic musculature in *Sinovenator* (Fig. 6C) corresponds well with the arrangements in other theropods, including previous troodontid reconstructions (*Hutchinson, 2001a*; *Hutchinson, 2002*; *Hutchinson et al., 2008*; *Bishop et al., 2018a*). The relatively small pelvis, and relatively small body size, limit space available for the origins of many muscles compared to more basal theropods such as *Falcarius* (Fig. 6A) or caenagnathids (Fig. 6B). On the lateral side, the short preacetabular blade restricts origins of deep dorsal musculature, and the shallow postacetabular blade reduces origins of crural flexors (Fig. 6C). Medially, the cuppedicus and brevis fossae, which are both narrower and shallower, housed similarly reduced origins of M. puboischiofemoralis internus 2 and M. caudofemoralis brevis (Fig. 6C). Conversely, the transverse expansion of the pubic apron (*Xu et al., 2017*) formed a broad space for the origins of Mm. puboischiofemorales externi 1–2 (Fig. 6C). Scarring indicates that the origin of M. puboischiofemoralis externus 2 filled the embayment on the posterior side of the pubic apron, bordered by the mediolaterally narrow edges of the shafts (Fig. 6C). Musculature that arose from the ischium was generally diminished with reduction in the size of the osteological correlates (Fig. 6C).

Derived Troodontidae indet. Derived troodontids show a relative increase in the size of the pelvis compared to more basal forms, and spaces for muscle attachment tend to track osteological changes within Troodontidae (Fig. 6D). Expansion of the postacetabulum, anteroposteriorly and dorsoventrally, also enlarges the origins of M. iliofibularis and M. flexor tibialis externus on the lateral side (Fig. 6D). Medially, the brevis fossa— and by extension, the origin of M. caudofemoralis brevis—was likewise expanded both dorsoventrally and mediolaterally (Fig. 6D). On the pubis, the hypertrophied preacetabular tubercle appears to also increase the origin of M. ambiens (Fig. 6D). The origin of M.

 

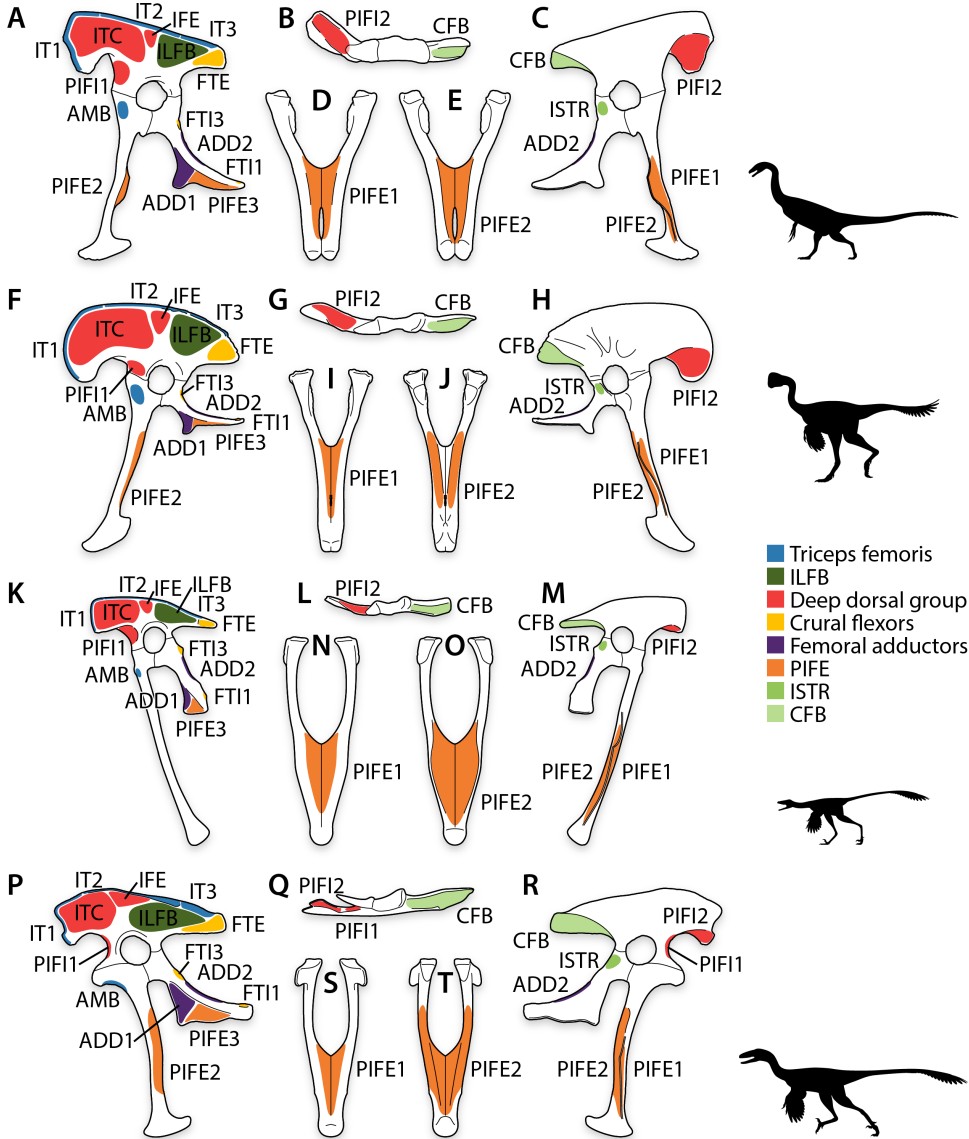

**Figure 6 Pelvic myology of non-avian maniraptorans.** Pelvis of *Falcarius* in left lateral view (A), ilium in ventral view (B), pelvis in medial view (C), and pubes in anterior (D) and posterior (E) views. Pelvis of Caenagnathidae indet. in left lateral view (F), ilium in ventral view (G), pelvis in medial view (H), and pubes in anterior (I) and posterior (J) views. Pelvis of *Sinovenator* in left lateral view (K), ilium in ventral view (L), pelvis in medial view (M), and pubes in anterior (N) and posterior (O) views. Pelvis of derived Troodontidae indet. in left lateral view (P), ilium in ventral view (Q), pelvis in medial view (R), and pubes in anterior (S) and posterior (T) views. See Table 2 for muscle abbreviations.

puboischiofemoralis externus 2 envelops the entire pubic apron and posterior sides of the pubic shafts, which is visible in lateral view (Fig. 6D). The scarred region indicates that this origin extends proximally halfway between the top of the apron and the ischiadic peduncle, filling a shallow longitudinal depression on the medial side of the shaft (Figs. 5C, 5L–5M, 6D), and distally to near the pubic boot (Fig. 6D). This differs considerably from

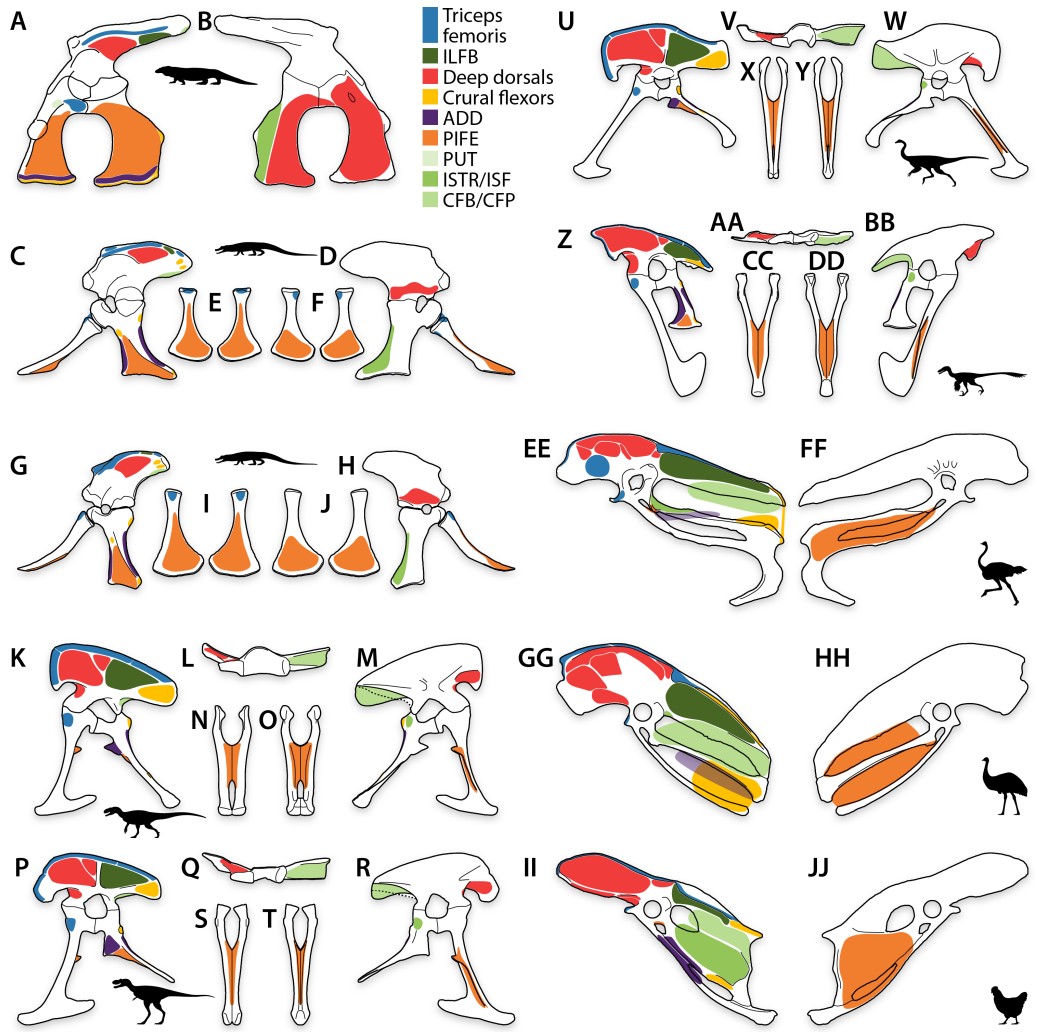

**Figure 7** **Pelvic myology of other study taxa.** Pelvis of *Varanus* in left lateral (A) and medial (B) views. Pelvis of *Alligator* in left lateral (C) and medial (D) views, and pubes in dorsal (E) and ventral (F) views. Pelvis of *Caiman* in left lateral (G) and medial (H) views, and pubes in dorsal (I) and ventral (J) views. Pelvis of *Allosaurus* in left lateral view (K), ilium in ventral view (L), pelvis in medial view (M), and pubes in anterior (N) and posterior (O) views. Pelvis of *Albertosaurus* in left lateral view (P), ilium in ventral view (Q), pelvis in medial view (R), and pubes in anterior (S) and posterior (T) views. Pelvis of Ornithomimidae indet. in left lateral view (U), ilium in ventral view (V), pelvis in medial view (W), and pubes in anterior (X) and posterior (Y) views. Pelvis of *Saurornitholestes* in left lateral view (Z), ilium in ventral view (AA), pelvis in medial view (BB), and pubes in anterior (CC) and posterior (DD) views. Pelvis of *Struthio* in left lateral (EE) and medial (FF) views. Pelvis of *Dromaius* in left lateral (GG) and medial (HH) views. Pelvis of *Gallus* in left lateral (II) and medial (JJ) views. See Table 2 for muscle abbreviations and refer to Supplemental Figures for individually labeled origins.

the condition reported in any other archosaur (*Hutchinson, 2001a*; *Hutchinson, 2002*), even from the condition identified in caenagnathids (Fig. 6B). In contrast, musculature on the ischium matches the layouts in other theropods (Figs. 7D–7G) (*Hutchinson, 2001a*; *Hutchinson, 2002*; *Bishop et al., 2018a*), although the enlarged obturator process increased the origins of M. adductor femoris 1 and M. puboischiofemoralis externus 3 (Fig. 6D).

## Area of attachment

The area of each muscle attachment was quantified (Table 3) and then calculated as a proportion of the total area of pelvic muscle origins (Table 4). This permits comparison of conventional anatomical groups, individual muscles, and functional groups that form antagonistic pairs (e.g., hip flexors and extensors). Relative proportions were favoured over absolute area to normalize the area of individual muscles or groups to the total area of pelvic musculature.

Conventional anatomical muscle groups exhibit notable variation across the sampled taxa (Fig. 8A). The triceps femoris is a relatively small group, occupying 16% or less of the total area of pelvic muscle origins in any taxon (Table 4). The area of this group peaks in *Allosaurus* and steadily decreases crownward throughout theropods to Troodontidae (3.5–6%), whereas extant relatives remain around the middle of this range (3–9%). More variation is shown by the deep dorsal group, which is largest in *Varanus* (47%), dwarfing the same group in crocodylians (18–19%) (Fig. 8A). This area gradually rises back to 47% in Caenagnathidae, steadily falls to 22% in derived Troodontidae, and then hovers at 15–23% in birds (Table 4). The flexor cruris group is modest in all taxa (1–9%) except for *Dromaius*, which has an exceptionally large area devoted to this group (15%). Fluctuation occurs across taxa for the external puboischiofemoral group (Fig. 8A). Although large in *Varanus* (34%) and predominant in crocodylians (52–60%), the external puboischiofemoral musculature drops in non-maniraptoran Theropoda (11–16%), spikes in non-avian Maniraptora (18–49%), and drops again in birds (27–34%). In particular, the Mm. puboischiofemorales externi are proportionally largest in *Sinovenator* (Table 4). Contrasting this pattern, the femoral adductor musculature hovers between 2–7% across all taxa (Fig. 8A). The remaining area is occupied by miscellaneous musculature, which is relatively small in squamates and crocodylians (7–13%), of moderate size in non-avian Maniraptora (25–42%), and rather substantial in non-maniraptoran Theropoda (48–52%) and in birds (36–54%).

Individual muscles also vary across sampled taxa, although some patterns emerge (Fig. 8B). Over half of the area of attachment of the triceps femoris group is comprised of the M. iliotibialis (and its homologues) in nearly every taxon except for *Struthio*, which instead is dominated by the origin of M. ambiens 2 (Fig. 8B). This is juxtaposed with other birds; the origin of M. ambiens occupies a small fraction of triceps femoris group and less than 0.2% of total pelvic muscle origin area in either *Dromaius* or *Gallus* (Table 4; Fig. 8B). Comprising the deep dorsal group, derivatives of the M. iliofemoralis and Mm. puboischiofemorales interni are generally similar among most taxa. However, it consists primarily of the Mm. puboischiofemorales interni in *Varanus*, a pattern reversed in *Gallus* by the overwhelmingly large origin of M. iliotrochantericus caudalis (Fig. 8B), an important hip flexor in extant birds (*Hutchinson & Gatesy, 2000*). The flexor cruris group is occupied mostly by the M. flexor tibialis externus in non-avian theropods, by the M. flexor cruris medialis in palaeognaths, and by both of these muscles roughly equally in *Gallus* (Fig. 8B). The external puboischiofemoral and femoral adductor groups remain similar across all taxa. However, the M. obturatorius medialis became the dominant muscle in birds after a protracted but steady increase in proportional area spanning

Rhodes et al. (2021), *PeerJ*, DOI 10.7717/peerj.10855

**Table 3** **Individual pelvic muscle origin areas (cm$^2$) for each taxon measured in both programs (Corel DRAW!, ImageJ).** Numbered columns correspond to taxa in Fig. 1: 1, *Varanus*; 2, *Alligator*; 3, *Caiman*; 4, *Allosaurus*; 5, *Albertosaurus*; 6, Ornithomimidae indet.; 7, *Falcarius*; 8, Caenagnathidae indet.; 9, *Saurornitholestes*; 10, *Sinovenator*; 11, derived Troodontidae indet.; 12, *Struthio*; 13, *Dromaius*; 14, *Gallus*.

| Muscle | 1 | | 2 | 3 | | 4 | 5 | 6 | 7 | 8 | 9 | 10 | 11 | | 12 | 13 | 14 |
|---|---|---|---|---|---|---|---|---|---|---|---|---|---|---|---|---|---|
| IT | 1.6400, 1.5928 | IT1 | 0.3517, 0.3542 | 0.1454, 0.1430 | IT1 | 55.5300, 52.8585 | 46.2600, 42.0757 | 32.0000, 30.8872 | 9.9360, 9.9736 | 7.0950, 7.6908 | 1.6960, 1.6466 | 0.1864, 0.1846 | 5.9820, 5.9757 | IC | 5.2640, 4.7024 | 5.4600, 5.2366 | 0.1313, 0.0982 |
| | | IT2 | 2.4520, 2.4262 | 0.4741, 0.4748 | IT2 | 97.1300, 91.6356 | 106.9600, 95.9770 | 40.4900, 36.5978 | 7.6660, 7.6134 | 4.7660, 4.8012 | 2.4310, 2.3680 | 0.2227, 0.2045 | 8.8970, 7.8217 | IL | 19.1300, 17.2724 | 8.7410, 7.9932 | 0.9124, 0.8003 |
| | | IT3 | 0.2799, 0.2809 | 0.0444, 0.0451 | IT3 | 57.5700, 54.0924 | 50.0000, 47.1902 | 20.0600, 17.8012 | 3.9640, 3.9783 | 1.6730, 1.5734 | 1.8130, 1.8660 | 0.1614, 0.1576 | 8.0940, 8.1602 | | | | |
| AMB | 0.9300, 0.9248 | AMB1 | 1.4922, 1.4224 | 0.1474, 0.2983 | AMB | 26.2600, 26.3178 | 33.0100, 33.5779 | 7.7840, 7.9078 | 4.4920, 4.6759 | 5.6800, 5.7987 | 1.5420, 1.5525 | 0.1071, 0.1054 | 3.4340, 3.1558 | AMB1 | 4.6970, 4.5010 | 0.5416, 0.5251 | 0.0476, 0.0337 |
| | | AMB2 | 0.5623, 0.5528 | a | | | | | | | | | | AMB2 | 33.9200, 33.3790 | a | a |
| ILFB | 0.9831, 0.9917 | ILFB | 0.4943, 0.4898 | 0.0584, 0.0582 | ILFB | 220.6500, 219.0626 | 286.4300, 284.1450 | 139.2000, 140.4796 | 31.1600, 31.1771 | 38.2600, 38.6444 | 8.6000, 8.6188 | 1.6010, 1.5939 | 60.7500, 62.4600 | ILFB | 124.5000, 121.6905 | 54.2400, 53.7954 | 1.0200, 1.0209 |
| IF | 3.2590, 3.2196 | IF | 4.9550, 4.9495 | 1.2320, 1.2239 | ITC | 250.8700, 251.2287 | 326.9200, 327.2630 | 203.1000, 205.1997 | 68.9700, 68.9311 | 88.5700, 89.4307 | 25.1900, 25.2433 | 3.3510, 3.3261 | 66.1900, 68.1188 | ITC | 68.2400, 66.9022 | 23.7000, 23.4878 | 5.2350, 5.2990 |
| | | | | | IFE | 40.9300, 40.4700 | 59.4800, 59.3447 | 33.2800, 33.7780 | 4.7840, 4.8695 | 8.5820, 8.8027 | 1.1360, 1.1367 | 0.3874, 0.3918 | 14.5600, 14.9618 | IFE | 3.5170, 3.4440 | 2.7120, 2.6996 | 0.2542, 0.2562 |
| PIFI1+2 | 12.4400, 12.3371 | PIFI1 | 5.0000, 5.0035 | 1.0570, 1.0508 | PIFI1 | 32.7620, 29.4096 | 40.1300, 38.7543 | 10.2700, 10.4166 | 9.0860, 9.2023 | 5.2260, 5.3039 | 3.8130, 3.8179 | 0.5818, 0.5629 | 7.8920, 7.1780 | IFI | 24.8800, 24.5188 | 10.6000, 10.5831 | 0.1607, 0.1585 |
| PIFI3 | 17.4800, 17.2989 | PIFI2 | b | b | PIFI2 | 76.7100, 74.2587 | 139.6400, 139.4410 | 37.2570, 34.4643 | 48.2600, 48.4686 | 36.9500, 37.2976 | 6.0878, 6.0420 | 0.6499, 0.6326 | 12.8270, 12.6053 | ITCR | 10.4200, 10.2395 | 17.7500, 17.8404 | 0.3324, 0.3246 |
| | | | | | | | | | | | | | | ITM | 5.1550, 5.1196 | 14.5300, 14.4034 | 0.1229, 0.1167 |
| PIT1 | 1.0703, 1.0527 | PIT | 0.3484, 0.3562 | 0.0437, 0.0424 | | | | | | | | | | | | | |
| FTI1 | 0.0975, 0.0917 | FTI1 | 0.1089, 0.1025 | 0.0537, 0.0477 | FTI1 | 4.4580, 1.7834 | 2.6000, 1.1192 | 1.3600, 1.1719 | 0.2051, 0.2206 | 0.1120, 0.1002 | 0.2715, 0.2738 | 0.0380, 0.0336 | 0.9766, 0.8363 | | | | |
| FTI2 | 0.1463, 0.1412 | FTI3 | 0.2941, 0.2876 | 0.0714, 0.0617 | FTI3 | 10.8050, 9.3902 | 8.4900, 6.7431 | 3.8110, 3.1756 | 0.5151, 0.4786 | 0.4788, 0.4205 | 0.2018, 0.2147 | 0.0853, 0.0815 | 2.0560, 1.9346 | FCM | 33.1700, 32.7531 | 60.5900, 59.8530 | 0.3248, 0.3309 |
| | | FTI2 | 0.2269, 0.2242 | 0.0617, 0.0627 | | | | | | | | | | | | | |
| FTE | b | FTE | 0.3097, 0.3133 | 0.0654, 0.0664 | FTE | 109.1400, 108.7883 | 87.4100, 87.6565 | 56.0700, 56.7627 | 9.9390, 10.0234 | 11.5200, 11.7734 | 1.3100, 1.3011 | 0.2524, 0.2533 | 14.8400, 15.2328 | FCLP | 11.5400, 10.8545 | 5.1080, 4.1161 | 0.3749, 0.3812 |
| PUT | 0.2832, 0.2829 | | | | | | | | | | | | | | | | |
| ADD | 2.9480, 2.9011 | ADD1 | 1.1800, 1.1320 | 0.1945, 0.1620 | ADD1 | 21.4400, 20.3833 | 70.1600, 70.5402 | 11.2000, 11.4542 | 11.4900, 11.3838 | 4.3730, 4.4071 | 3.8960, 3.8026 | 0.3297, 0.3199 | 18.5000, 18.7969 | PIFM | 34.6600, 33.6699 | 32.2500, 31.2045 | 0.7629, 0.7502 |
| | | ADD2 | 2.1050, 1.9424 | 0.3689, 0.3533 | ADD2 | 15.5780, 11.3672 | 20.9200, 15.7738 | 5.4710, 4.0141 | 3.5520, 3.2151 | 2.1880, 1.7322 | 1.6358, 1.6140 | 0.2719, 0.2437 | 8.7980, 7.1172 | PIFL | a | a | 0.5581, 0.5577 |
| PIFE | 23.9870, 23.7727 | PIFE1 | 11.0530, 10.7369 | 3.2169, 3.1587 | PIFE1 | 72.7600, 67.0230 | 127.7400, 112.7633 | 39.2900, 35.5202 | 38.6400, 37.2324 | 19.8090, 18.5156 | 8.3580, 7.8680 | 3.1035, 2.8996 | 27.1930, 25.7227 | OL | 5.5530, 4.2878 | – | 0.0550, 0.0479 |
| | | PIFE2 | 8.5360, 8.3300 | 2.5499, 2.5081 | PIFE2 | 107.0220, 100.7226 | 136.6600, 123.9844 | 49.8200, 46.7320 | 32.3490, 29.5996 | 31.6290, 31.3789 | 9.9870, 8.7183 | 6.0005, 5.6255 | 107.3300, 104.8976 | OM | 219.7000, 213.6802 | 118.2600, 116.7258 | 9.0040, 9.1144 |
| | | PIFE3 | 8.7910, 8.5178 | 1.5350, 1.5155 | PIFE3 | 4.5720, 3.8392 | 43.8100, 42.4911 | 10.3600, 9.3188 | 9.5260, 9.4732 | 2.8430, 2.7976 | 2.4950, 2.5057 | 0.4311, 0.4234 | 17.3000, 17.6725 | | | | |
| ISTR | 4.5300, 4.4808 | ISTR | 5.1060, 4.9072 | 0.6896, 0.6421 | ISTR | 13.8900, 14.1762 | 30.2600, 30.4392 | 5.2110, 5.3750 | 3.2980, 3.3710 | 1.8270, 1.8755 | 1.2260, 1.2374 | 0.1784, 0.1767 | 5.3410, 5.6510 | ISF | 18.9800, 17.2033 | 0.8023, 0.7917 | 4.9530, 5.0159 |
| CFB | 0.1671, 0.1254 | CFB | 0.8096, 0.7836 | 0.1268, 0.1245 | CFB | 250.0600, 230.2807 | 335.7100, 314.0476 | 164.4500, 157.6395 | 18.2660, 17.7387 | 23.4300, 23.7332 | 10.2510, 10.1139 | 1.6092, 1.5163 | 64.0600, 62.0356 | CFP | 141.4000, 138.6220 | 90.1900, 89.0167 | 2.6540, 2.6837 |

**Notes.**

[a] undivided.

[b] origin located on soft tissue (not directly on pelvis).

See Table 2 for muscle abbreviations.

**Table 4 Relative proportion of individual origin area to total area of all origins for each taxon in both programs (Corel DRAW!, ImageJ).** Numbered columns correspond to taxa in Fig. 1: 1, *Varanus*; 2, *Alligator*; 3, *Caiman*; 4, *Allosaurus*; 5, *Albertosaurus*; 6, Ornithomimidae indet.; 7, *Falcarius*; 8, Caenagnathidae indet.; 9, *Saurornitholestes*; 10, *Sinovenator*; 11, derived Troodontidae indet.; 12, *Struthio*; 13, *Dromaius*; 14, *Gallus*.

| Muscle | 1 | Muscle | 2 | 3 | Muscle | 4 | 5 | 6 | 7 | 8 | 9 | 10 | 11 | Muscle | 12 | 13 | 14 |
|---|---|---|---|---|---|---|---|---|---|---|---|---|---|---|---|---|---|
| IT | 0.0234, 0.0230 | IT1 | 0.0065, 0.0067 | 0.0120, 0.0119 | IT1 | 0.0378, 0.0376 | 0.0237, 0.0225 | 0.0368, 0.0364 | 0.0314, 0.0320 | 0.0240, 0.0260 | 0.0184, 0.0183 | 0.0095, 0.0099 | 0.0131, 0.0133 | IC | 0.0069, 0.0063 | 0.0123, 0.0119 | 0.0049, 0.0036 |
|  |  | IT2 | 0.0450, 0.0457 | 0.0391, 0.0394 | IT2 | 0.0662, 0.0651 | 0.0548, 0.0512 | 0.0465, 0.0431 | 0.0243, 0.0244 | 0.0162, 0.0162 | 0.0264, 0.0263 | 0.0114, 0.0109 | 0.0196, 0.0174 | IL | 0.0250, 0.0233 | 0.0196, 0.0182 | 0.0339, 0.0297 |
|  |  | IT3 | 0.0051, 0.0053 | 0.0037, 0.0037 | IT3 | 0.0392, 0.0384 | 0.0256, 0.0252 | 0.0230, 0.0210 | 0.0125, 0.0128 | 0.0057, 0.0053 | 0.0197, 0.0207 | 0.0083, 0.0084 | 0.0178, 0.0181 |  |  |  |  |
| AMB | 0.0133, 0.0134 | AMB1 | 0.0274, 0.0268 | 0.0121, 0.0248 | AMB | 0.0179, 0.0187 | 0.0169, 0.0179 | 0.0089, 0.0093 | 0.0142, 0.0150 | 0.0193, 0.0196 | 0.0168, 0.0173 | 0.0055, 0.0056 | 0.0075, 0.0070 | AMB1 | 0.0061, 0.0061 | 0.0012, 0.0012 | 0.0018, 0.0012 |
|  |  | AMB2 | 0.0103, 0.0104 | [a] |  |  |  |  |  |  |  |  |  | AMB2 | 0.0444, 0.0449 | [a] | [a] |
| ILFB | 0.0141, 0.0143 | ILFB | 0.0091, 0.0092 | 0.0048, 0.0048 | ILFB | 0.1503, 0.1557 | 0.1467, 0.1517 | 0.1599, 0.1655 | 0.0986, 0.1000 | 0.1297, 0.1305 | 0.0935, 0.0958 | 0.0819, 0.0851 | 0.1335, 0.1387 | ILFB | 0.1628, 0.1638 | 0.1218, 0.1227 | 0.0379, 0.0378 |
| IF | 0.0466, 0.0465 | IF | 0.0910, 0.0932 | 0.1015, 0.1017 | ITC | 0.1709, 0.1785 | 0.1674, 0.1747 | 0.2333, 0.2418 | 0.2182, 0.2212 | 0.3002, 0.3021 | 0.2740, 0.2807 | 0.1714, 0.1776 | 0.1455, 0.1513 | ITC | 0.0892, 0.0901 | 0.0532, 0.0536 | 0.1946, 0.1963 |
|  |  |  |  |  | IFE | 0.0279, 0.0288 | 0.0305, 0.0317 | 0.0382, 0.0398 | 0.0151, 0.0156 | 0.0291, 0.0297 | 0.0124, 0.0126 | 0.0198, 0.0209 | 0.0320, 0.0332 | IFE | 0.0046, 0.0046 | 0.0061, 0.0062 | 0.0094, 0.0095 |
| PIFI1+2 | 0.1778, 0.1782 | PIFI1 | 0.0918, 0.0942 | 0.0871, 0.0873 | PIFI1 | 0.0223, 0.0209 | 0.0206, 0.0207 | 0.0118, 0.0123 | 0.0287, 0.0295 | 0.0177, 0.0179 | 0.0415, 0.0424 | 0.0298, 0.0301 | 0.0173, 0.0159 | IFI | 0.0325, 0.0330 | 0.0238, 0.0241 | 0.0060, 0.0059 |
| PIFI3 | 0.2499, 0.2499 | PIFI2 | [b] | [b] | PIFI2 | 0.0522, 0.0528 | 0.0715, 0.0744 | 0.0428, 0.0406 | 0.1527, 0.1555 | 0.1252, 0.1260 | 0.0662, 0.0672 | 0.0332, 0.0338 | 0.0282, 0.0280 | ITCR | 0.0136, 0.0138 | 0.0398, 0.0407 | 0.0124, 0.0120 |
|  |  |  |  |  |  |  |  |  |  |  |  |  |  | ITM | 0.0067, 0.0069 | 0.0326, 0.0329 | 0.0046, 0.0043 |
| PIT1 | 0.0153, 0.0152 | PIT | 0.0064, 0.0067 | 0.0036, 0.0035 |  |  |  |  |  |  |  |  |  |  |  |  |  |
| FTI1 | 0.0014, 0.0013 | FTI1 | 0.0020, 0.0019 | 0.0044, 0.0040 | FTI1 | 0.0030, 0.0013 | 0.0013, 0.0006 | 0.0016, 0.0014 | 0.0006, 0.0007 | 0.0004, 0.0003 | 0.0030, 0.0030 | 0.0019, 0.0018 | 0.0021, 0.0019 |  |  |  |  |
| FTI2 | 0.0021, 0.0020 | FTI3 | 0.0054, 0.0054 | 0.0059, 0.0051 | FTI3 | 0.0074, 0.0067 | 0.0043, 0.0036 | 0.0044, 0.0037 | 0.0016, 0.0015 | 0.0016, 0.0014 | 0.0022, 0.0024 | 0.0044, 0.0044 | 0.0045, 0.0043 | FCM | 0.0434, 0.0441 | 0.1360, 0.1366 | 0.0121, 0.0123 |
|  |  | FTI2 | 0.0042, 0.0042 | 0.0051, 0.0052 |  |  |  |  |  |  |  |  |  |  |  |  |  |
| FTE | [b] | FTE | 0.0057, 0.0059 | 0.0054, 0.0055 | FTE | 0.0743, 0.0773 | 0.0448, 0.0468 | 0.0644, 0.0669 | 0.0314, 0.0322 | 0.0390, 0.0398 | 0.0142, 0.0145 | 0.0129, 0.0135 | 0.0326, 0.0338 | FCLP | 0.0151, 0.0146 | 0.0115, 0.0094 | 0.0139, 0.0141 |
| PUT | 0.0040, 0.0041 |  |  |  |  |  |  |  |  |  |  |  |  |  |  |  |  |
| ADD | 0.0421, 0.0419 | ADD1 | 0.0217, 0.0213 | 0.0160, 0.0135 | ADD1 | 0.0146, 0.0145 | 0.0359, 0.0377 | 0.0129, 0.0135 | 0.0363, 0.0365 | 0.0148, 0.0149 | 0.0424, 0.0423 | 0.0169, 0.0171 | 0.0407, 0.0417 | PIFM | 0.0453, 0.0453 | 0.0724, 0.0712 | 0.0284, 0.0278 |
|  |  | ADD2 | 0.0387, 0.0366 | 0.0304, 0.0293 | ADD2 | 0.0106, 0.0081 | 0.0107, 0.0084 | 0.0063, 0.0047 | 0.0112, 0.0103 | 0.0074, 0.0059 | 0.0178, 0.0179 | 0.0139, 0.0130 | 0.0193, 0.0158 | PIFL | [a] | [a] | 0.0207, 0.0207 |
| PIFE | 0.3429, 0.3435 | PIFE1 | 0.2030, 0.2022 | 0.2651, 0.2624 | PIFE1 | 0.0496, 0.0476 | 0.0654, 0.0602 | 0.0451, 0.0419 | 0.1222, 0.1195 | 0.0671, 0.0625 | 0.0909, 0.0875 | 0.1588, 0.1548 | 0.0598, 0.0571 | OL | 0.0073, 0.0058 | – | 0.0020, 0.0018 |
|  |  | PIFE2 | 0.1568, 0.1568 | 0.2101, 0.2083 | PIFE2 | 0.0729, 0.0716 | 0.0700, 0.0662 | 0.0572, 0.0551 | 0.1023, 0.0950 | 0.1072, 0.1060 | 0.1086, 0.0969 | 0.3070, 0.3003 | 0.2359, 0.2329 | OM | 0.2873, 0.2877 | 0.2655, 0.2663 | 0.3347, 0.3377 |
|  |  | PIFE3 | 0.1614, 0.1604 | 0.1265, 0.1259 | PIFE3 | 0.0031, 0.0027 | 0.0224, 0.0227 | 0.0119, 0.0110 | 0.0301, 0.0304 | 0.0096, 0.0094 | 0.0271, 0.0279 | 0.0221, 0.0226 | 0.0380, 0.0392 |  |  |  |  |
| ISTR | 0.0647, 0.0647 | ISTR | 0.0938, 0.0924 | 0.0568, 0.0533 | ISTR | 0.0095, 0.0101 | 0.0155, 0.0162 | 0.0060, 0.0063 | 0.0104, 0.0108 | 0.0062, 0.0063 | 0.0133, 0.0138 | 0.0091, 0.0094 | 0.0117, 0.0125 | ISF | 0.0248, 0.0232 | 0.0018, 0.0018 | 0.1841, 0.1858 |
| CFB | 0.0024, 0.0018 | CFB | 0.0149, 0.0148 | 0.0104, 0.0103 | CFB | 0.1703, 0.1637 | 0.1719, 0.1676 | 0.1889, 0.1857 | 0.0578, 0.0569 | 0.0794, 0.0802 | 0.1115, 0.1125 | 0.0823, 0.0809 | 0.1408, 0.1378 | CFP | 0.1849, 0.1866 | 0.2025, 0.2031 | 0.0987, 0.0994 |

**Notes.**

[a] undivided.

[b] origin located on soft tissue (not directly on pelvis).

See Table 2 for muscle abbreviations.

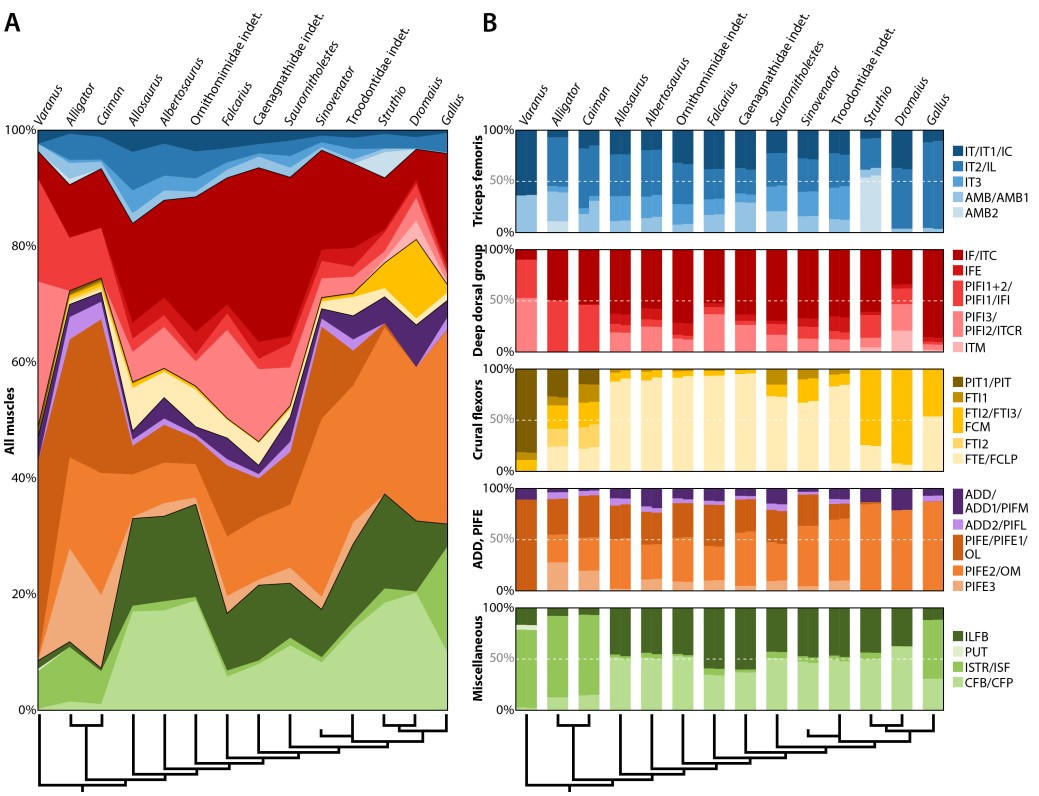

**Figure 8** **Area of attachment across all pelvic muscles and by conventional anatomical groups.** (A) Equal-area chart showing the proportion of individual muscles to the total area of all muscles. (B) Paired bar charts representing anatomical groups with results from Corel DRAW! (left bars) and ImageJ (right bars). See Table 2 for muscle abbreviations.

Archosauria (Fig. 8B). Among miscellaneous muscles, the M. ischiotrochantericus has the largest relative origin in extant saurians and *Gallus* (Table 4). This starkly contrasts with non-avian theropods and palaeognaths, in which the area of the M. ischiotrochantericus is eclipsed by the M. caudofemoralis brevis and M. iliofibularis, each accounting for nearly half of all miscellaneous musculature (Fig. 8B). Interestingly, the bulk of taxa examined (crocodylians, non-avian theropods, and palaeognaths) have greater numbers of muscles with osteological correlates on the pelvis than *Varanus* and *Gallus* at either end (Fig. 8B). In *Varanus*, this is due to many muscles remaining undivided in squamates relative to other taxa (Table 2). In *Gallus*, this is because some muscles have merged or were lost entirely (Table 2). Additionally, many groups in *Gallus* appear simplified because a single muscle dominates each of them: the M. iliotibialis lateralis in the triceps femoris, the M. iliotrochantericus caudalis among the deep dorsals, and the M. obturatorius medialis for the obturator muscles (Fig. 8B).

Categorizing pelvic muscles according to functional groups revealed a largely conservative pattern among non-avian theropods (Table 5). Even more broadly, this identified muscles involved in the same function across all taxa studied, such as the major extensors comprising parts of the triceps femoris (squamate M. iliotibialis, crocodylian

Mm. iliotibiales 2–3, and avian M. iliotibialis lateralis), M. iliofibularis, crural flexors, and M. caudofemoralis brevis. Yet, there is considerable variation across taxa when functions are compared in antagonistic pairs (Fig. 9A). The sum of origin areas for muscles that flex (protract) the leg around the hip joint exceed that for muscles that extend (retract) the leg in squamates and crocodylians. Non-maniraptoran theropods, alongside *Saurornitholestes* and derived troodontids, maintain a nearly 50:50 split between hip flexors and extensors, whereas *Falcarius*, the caenagnathid, *Sinovenator*, and *Gallus* have relatively more area devoted to hip flexion (Fig. 9A). The palaeognaths included here, *Struthio* and *Dromaius*, both have more pelvic area for extensors than any other taxon (Fig. 9A), mirroring findings of previous studies noting the adaptations for running in the musculature of these birds (*Patak & Baldwin, 1998*; *Smith et al., 2006*; *Smith et al., 2007*; *Lamas, Main & Hutchinson, 2014*; *Hutchinson et al., 2015*). Origin areas for musculature that adduct the hind limbs greatly outsize those that perform abduction about the hips in squamates and crocodylians (Fig. 9A). Most theropods seem to favour abduction instead, with the troodontids and *Gallus* as the only exceptions. This trend is mostly consistent with previous findings (*Hutchinson & Gatesy, 2000*), albeit measured using origin areas rather than other methods. Concerning long axis rotation, the pelves of *Varanus* and all three birds exhibit more area for lateral than medial rotators (Fig. 9A). Crocodylians and non-maniraptoran theropods hover around an even proportion of each, whereas non-avian maniraptorans appear to favour medial rotators. Lastly, muscles involved in knee flexion and extension are about equal in *Varanus* (Fig. 9A). Crocodylians have knee extensors with origins that cover much more area than flexors, which is the opposite case in all theropods.

Pruning these charts to remove novel muscle reconstructions, using only *Alligator* and *Struthio* as traditional extant representatives, show clearer patterns of locomotor evolution that more closely resemble the stepwise changes in function affirmed in previous studies (*Gatesy, 1990*; *Gatesy, 1995*; *Gatesy & Dial, 1996*; *Hutchinson & Gatesy, 2000*; *Hutchinson, 2002*; *Hutchinson & Allen, 2009*). Hip flexors and extensors are weighted more heavily toward flexion in *Alligator* but are nearly equal across non-avian theropods and *Struthio* (Fig. 9B). Hip abduction and adduction appears less straightforward, although still shows that a large area of pelvic muscle origins involved in abduction appeared early in theropod evolution (*Hutchinson & Gatesy, 2000*). Long axis rotation is not dramatically different between *Alligator* and non-maniraptoran theropods. However, the increase in area for medial rotators in *Saurornitholestes* suggests a pattern of increased lateral rotation on the line to birds (Fig. 9A). Around the knee, the majority of musculature is dedicated to extension in *Alligator*. This changes to flexion in *Allosaurus* and continues in a stepwise fashion to derived troodontids and birds, with *Saurornitholestes* as a single exception (Fig. 9B). Whereas the proportional areas of muscle groups vary considerably across taxa (Fig. 9, Table 4), functional groups tend to remain conservative (Fig. 10, Table 5). The inclusion of more data with novel reconstructions of maniraptoran pelvic musculature shows a more complicated pattern in the evolution of functional groups.

The origin areas derived from Corel DRAW! and ImageJ yielded similar results. None of the statistical tests showed evidence of significant differences in estimated muscle origin areas, $t(506) = 0.1723$, $p = 0.8633$ (Table 6). Comparison of the matrices produced by

Rhodes et al. (2021), *PeerJ*, DOI 10.7717/peerj.10855

Peer J

**Table 5 Inferred muscle functions around the hip (h) and knee (k) joints including flexion (F), extension (E), abduction (Ab), adduction (Ad), lateral rotation (L), and medial rotation (M).** Functions are averaged for each muscle and are derived from previous studies cited in the main text. Numbered columns correspond to taxa in Fig. 1: 1, *Varanus*; 2, *Alligator*; 3, *Caiman*; 4, *Allosaurus*; 5, *Albertosaurus*; 6, Ornithomimidae indet.; 7, *Falcarius*; 8, Caenagnathidae indet.; 9, *Saurornitholestes*; 10, *Sinovenator*; 11, derived Troodontidae indet.; 12, *Struthio*; 13, *Dromaius*; 14, *Gallus*.

| Muscle | 1 | | 2–3 | | 4 | 5 | 6–8 | 9–11 | | 12 | 13 | 14 |
|---|---|---|---|---|---|---|---|---|---|---|---|---|
| IT | hE, Ab; kE | IT1 | hF, Ab; kE | IT1 | hF, Ab, M; kE | hF, Ab, M; kE | hF, Ab, M; kE | hF, Ab, M; kE | IC | hF, Ad, M, kF/E | hF, Ad, M, kF/E | hF; kE |
| | | IT2 | hF/E, Ab; kE | IT2 | hF/E, Ab, M/L; kE | hF/E, Ab, M/L; kE | hF/E, Ab, M/L; kE | hF/E, Ab, M/L; kE | IL | hF/E, Ab, M/L; kE | hE, Ab, M/L; kE | hF/E, Ab; kE |
| | | IT3 | hE, Ab, L; kE | IT3 | hE, Ab, L; kE | hE, Ab, L; kE | hE, Ab, L; kE | hE, Ab, L; kE | | | | |
| AMB | kE | AMB1 | hF/E, M; kE | AMB | hF/E, M; kE | hF/E, M; kE | hF/E, M; kE | hF/E, M; kE | AMB1 | hF/E, Ad, L; kF | hAd, M; kF | hF, Ad; kE |
| | | AMB2 | hF/E, M; kE | | | | | | AMB2 | hF, Ad, M/L; kE | a | a |
| ILFB | hE, Ab; kF | ILFB | hE, Ab, L; kF | ILFB | hE, Ab, L; kF | hE, Ab, L; kF | hE, Ab, L; kF | hE, Ab, L; kF | ILFB | hE, Ab, M; kF | hE, Ab; kF | hE, Ab; kF |
| IF | hE, Ab | IF | hF/E, Ab, L | ITC | hF, Ab, M | hF, Ab, M | hF, Ab, M | hF/E, Ab, M | ITC | hF/E, Ab/Ad, M | hF, Ab/Ad, M | hF, Ab, M |
| | | | | IFE | hF/E, Ab, L | hF/E, Ab, L | hF/E, Ab, L | hF/E, Ab, L | IFE | hF, Ab, M/L | hF, Ab, M/L | hF, Ab |
| PIFI1+2 | hF, Ab, M | PIFI1 | hF, Ad | PIFI1 | hF, Ab, M | hF, Ab, M | hF, Ab, M | hF, Ab, M | IFI | hF, Ad, M/L | hF, Ad, M/L | hF, Ad |
| PIFI3 | hF, L | PIFI2 | b | PIFI2 | hF, Ab, M | hF, Ab, M | hF, Ab, M | hF/E, Ab, M | ITCR | hF/E, Ab/Ad, M | hF, Ab/Ad, M | hF, M |
| | | | | | | | | | ITM | hF/E, Ab/Ad, M | hF, Ab/Ad, M | hF, Ab, M |
| PIT1 | hF, Ad; kF | PIT | hE, Ad, M/L; kF | | | | | | | | | |
| FTI1 | hE, Ad; kF | FTI1 | hE, Ad, M/L; kF | FTI1 | hE, Ad, M/L; kF | hE, Ad, M/L; kF | hE, M/L; kF | hE, M/L; kF | | | | |
| FTI2 | hE, Ad; kF | FTI3 | hE, Ad; kF | FTI3 | hE; kF | hE; kF | hE, Ad; kF | hE, Ad; kF | FCM | hE, Ab, M; kF | hE, Ab, M; kF | hE, Ab; kF |
| | | FTI2 | ? | | | | | | | | | |
| FTE | b | FTE | hE, Ab, L; kF | FTE | hE, Ab; kF | hE, Ab; kF | hE, Ab; kF | hE, Ab; kF | FCLP | hE, Ab, M; kF | hE, Ab, M; kF | hE, Ab; kF |
| PUT | hF, Ad; kF | | | | | | | | | | | |
| ADD | hF, Ad | ADD1 | hF/E, Ad, M/L | ADD1 | hF/E, Ad, L | hF/E, Ad, L | hF/E, Ad, L | hE, Ad, L | PIFM | hE, Ab, L | hE, Ab, L | hE |
| | | ADD2 | hE, Ad, L | ADD2 | hF/E, Ad, L | hE, Ad, L | hE, Ad, L | hE, Ad, L | PIFL | a | a | hE |
| PIFE | hF/E, Ad, L | PIFE1 | hF, Ad, M | PIFE1 | hF, Ad, M | hF, Ad, M | hF, Ad, M | hF, Ad, M | OL | hF, Ad | – | hF, Ad, L |
| | | PIFE2 | hF, Ad, M | PIFE2 | hF, Ad, M | hF, Ad, M | hF, Ad, M | hF, Ad, M | OM | hF, Ab/Ad, L | hF, Ad, L | hF, Ad, L |
| | | PIFE3 | hF, Ad, L | PIFE3 | hF/E, Ad, L | hF/E, Ad, L | hF/E, Ad, L | hF/E, Ad, L | | | | |
| ISTR | hE, L | ISTR | hF/E, Ad, L | ISTR | hE, Ad, L | hE, Ad, L | hE, Ad, L | hE, Ad, L | ISF | hF/E, Ab, L | hF/E, Ab, L | hE, L |
| CFB | hE, M | CFB | hE, M | CFB | hE, L | hE, L | hE, L | hE, L | CFP | hE, Ab, L | hE, Ab, L | hE |

**Notes.**

[a] undivided.

[b] origin located on soft tissue (not directly on pelvis).

See Table 2 for muscle abbreviations.

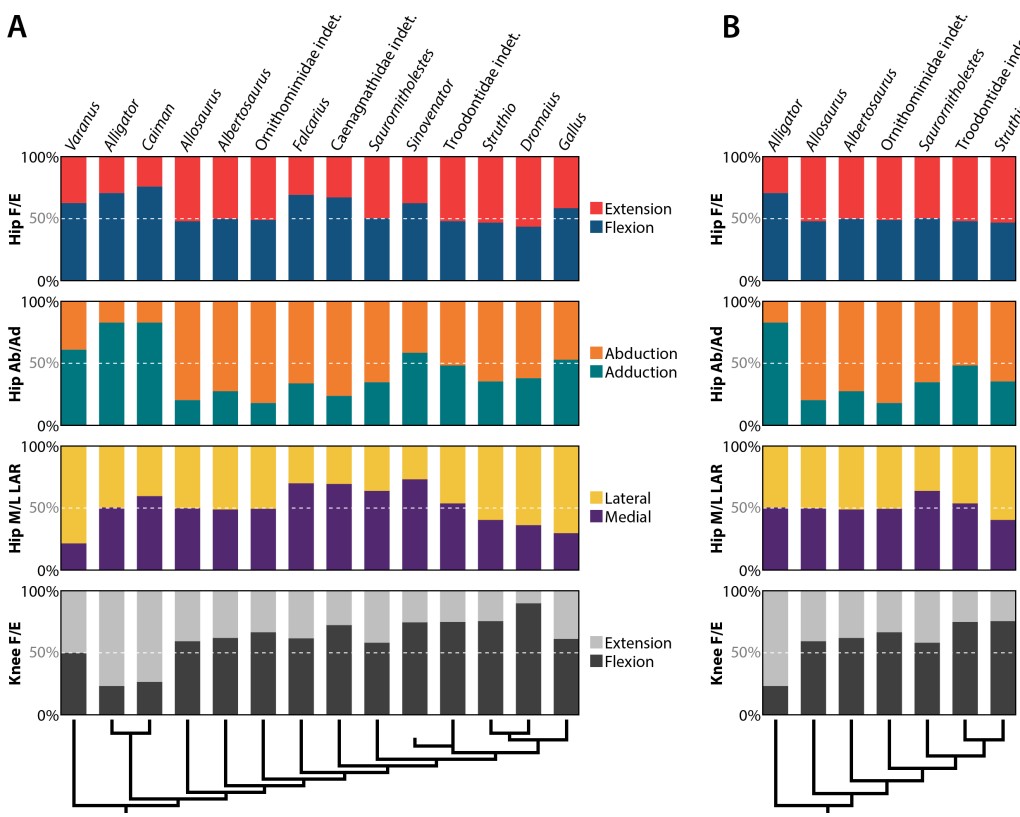

**Figure 9  Area of attachment by functional groups.** (A) Antagonistic pairs of all taxa sampled. (B) Antagonistic pairs of taxa pruned to reflect general results of past studies. Abbreviations: Ab, abduction; Ad, adduction; E, extension; F, flexion; M, medial; L, lateral; LAR, long axis rotation.

these two programs in a RV coefficients analysis (RV = 0.9989) further demonstrated the similarity of the measured origin areas (Table 6). Lastly, sensitivity analysis yielded virtually no differences among results measured using ImageJ under five tolerance levels (Fig. 11). This was true regardless of whether the analysis included only ImageJ results, $F(4, 90) = 1.006 \times 10^{-3}$, $p = 0.9999$, or included Corel DRAW! results as well, $F(5, 108) = 3.661 \times 10^{-3}$, $p = 0.9999$. Thus, method did not significantly affect the results of this study.

Body mass estimates for each taxon were compared to the area of attachment of all pelvic musculature, to the area of attachment of major extensors, and to ilium length (Table 7). These values were log-transformed and then subjected to Phylogenetic Generalized Least Squares (PGLS) regressions (Table 8). Regressions of all taxa demonstrated that phylogenetically corrected body size explained much but not all of the variation in the area of attachment for all hip muscles ($R^2_{adj} = 0.9095$) or major extensors ($R^2_{adj} = 0.8495$), or the variation in ilium length ($R^2_{adj} = 0.8677$). However, regressions of each major group (non-theropod Sauria, non-maniraptoran Theropoda, non-avian Maniraptora, and Aves) or only extinct taxa (non-avian Theropoda) tended to perform better (Table 8). The non-theropod saurians had notably different slopes and intercepts compared to all

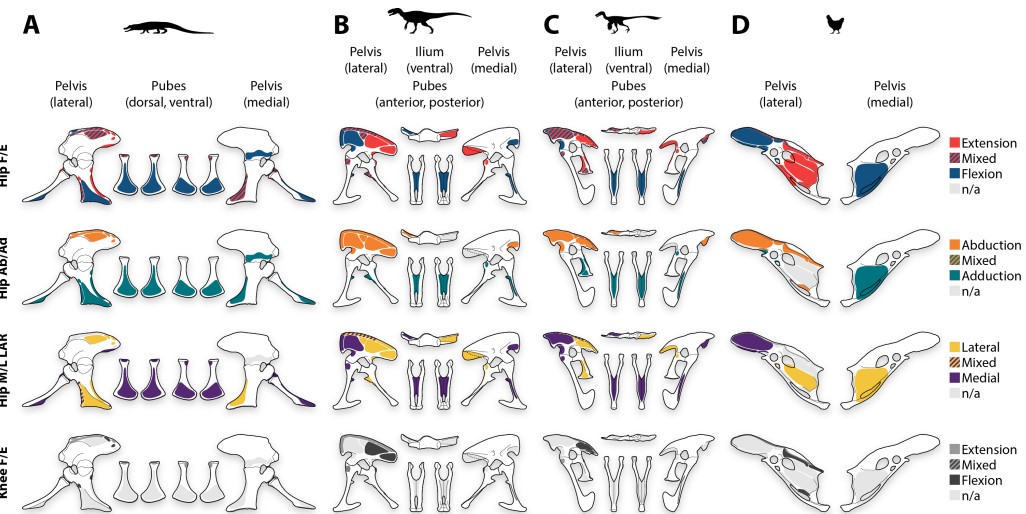

**Figure 10 Pelvic myology by functional groups.** Antagonistic pairs demonstrate general patterns in representatives of non-theropod Sauria (*Alligator*, A), non-maniraptoran Theropoda (*Allosaurus*, B), non-avian Maniraptora (*Saurornitholestes*, C), and Aves (*Gallus*, D). Abbreviations: Ab, abduction; Ad, adduction; E, extension; F, flexion; M, medial; L, lateral; LAR, long axis rotation.

**Table 6 Summary statistics of pelvic muscle origin areas (cm²) using two-sample *t*-tests assuming equal variances.**

| Group | $\bar{x}_{\text{CorelDRAW}}$ | $\bar{x}_{\text{ImageJ}}$ | *t*-value | df | *p*-value |
|---|---|---|---|---|---|
| *Varanus* | 4.9973 | 4.9438 | 0.0189 | 26 | 0.9850 |
| *Alligator* | 2.7228 | 2.6557 | 0.0639 | 38 | 0.9494 |
| *Caiman* | 0.6387 | 0.6336 | 0.0174 | 36 | 0.9862 |
| *Allosaurus* | 77.2704 | 74.0573 | 0.1251 | 36 | 0.9012 |
| *Albertosaurus* | 102.7679 | 98.5962 | 0.1255 | 36 | 0.9008 |
| Ornithomimdae indet. | 45.8149 | 44.6682 | 0.0608 | 36 | 0.9519 |
| *Falcarius* | 16.6367 | 16.4014 | 0.0389 | 36 | 0.9692 |
| Caenagnathidae indet. | 15.5269 | 15.5830 | −0.0080 | 36 | 0.9937 |
| *Saurornitholestes* | 4.8390 | 4.7338 | 0.0550 | 36 | 0.9564 |
| *Sinovenator* | 1.0289 | 0.9859 | 0.0875 | 36 | 0.9307 |
| Troodontidae indet. | 23.9485 | 23.7018 | 0.0263 | 36 | 0.9792 |
| *Struthio* | 44.9839 | 43.6965 | 0.0624 | 32 | 0.9506 |
| *Dromaius* | 29.6983 | 29.2182 | 0.0370 | 28 | 0.9708 |
| *Gallus* | 1.5825 | 1.5876 | −0.0059 | 32 | 0.9953 |
| All taxa | 26.9389 | 26.1350 | 0.1723 | 506 | 0.8633 |

**Notes.**

df, degrees of freedom; $\bar{x}$, mean.

other taxa, which is presumably related to the differences in posture between these two groups. This is because the regression lines for non-theropod saurians also represent the quadrupedal taxa examined. These data only include the pelvis, ignoring the shoulder and thus half of the quadrupedal locomotory apparatus, which is probably why the results

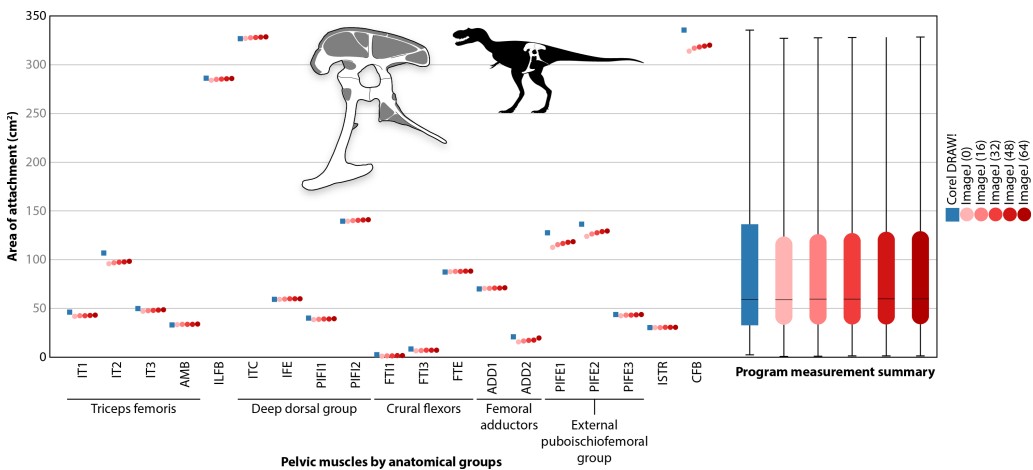

**Figure 11 Sensitivity analysis.** Comparison of Corel DRAW! and ImageJ under five tolerance levels performed on the *Albertosaurus* reconstruction (inset) shows no significant differences. Scatter plot (left) depicts individual muscle origin measurements, summarized in the box and whisker plot (right; coloured bars = interquartile range, black line = mean). See Table 2 for muscle abbreviations.

appear as outliers to the bipedal taxa. Regressions that only included the bipeds, or only non-avian theropods, tended to be better predictors than ones that included all taxa (Table 8).

The slopes of each regression reveal how much the dependent variable increases relative to an increase in body mass. However, all of the bivariate comparisons plot body mass (a three-dimensional property) against area (two-dimensional) or length (one-dimensional), which means scaling principles must be considered when comparing these values (*Biewener, 1989*; *Hutchinson & Garcia, 2002*; *Allen et al., 2010*; *Allen et al., 2015*). According to these principles, as body mass increases, area scales as (body mass)$^{2/3}$ and length scales as (body mass)$^{1/3}$. Because the regression lines can be expressed as the power equation y = bx$^k$ (*Currie, 2003*), the slope (k) can be directly compared to its respective scaling factor ($k_{area}$ = 2/3 [≈ 0.67]; $k_{length}$ = 1/3 [≈ 0.33]). Therefore, a slope near or equal to one ($k = 1$) indicates isometric growth, whereas a slope significantly greater than one (k > 1) signifies positive allometry and a slope significantly less than one (k < 1) denotes negative allometry. For non-theropod Sauria, the regression lines for area of attachment—both total hip muscles and just major extensors—have slopes about double what is expected based on the scaling factor of 0.67 (Table 8). In contrast, the slope for ilium length is close to 0.33. Non-maniraptoran theropods have slopes much lower than expected across all three comparisons (Table 8). This is likely due to the small sample size that contributes to lower $R^2_{adj}$ values, and the data point for Ornithomimidae indet. with relatively large areas of hip and extensor musculature for its estimated body mass, which skews the left side of this regression line. In non-avian maniraptorans, avians, non-avian theropods, and collectively among bipeds, the slopes for all three comparisons are reasonably close to their scaling factors, although the area of attachment of major extensors is somewhat higher than expected (Table 8).

**Table 7 Measurements of select pelvic elements, hip musculature, and limb elements or other sources used for body mass estimates (given to 1 significant digit).**

| Taxon | Specimen (ilium) | Length of ilium, anteroposterior (cm) | Area of all pelvic muscles (cm$^2$) | Area of major extensors (cm$^2$) | Specimen (femur, if not same as ilium) | Data source | Body mass estimate (kg) | Range (±25%) |
|---|---|---|---|---|---|---|---|---|
| *Varanus* | ROM R7565 | 9.8 | 70.0 | 4.0 | – | Specimen card | 45.0 | n/a |
| *Alligator* | ROM R343 | 9.0 | 54.5 | 6.8 | – | Calculated by ilium length (*Dodson, 1975*) | 43.8 | 32.8–54.7 |
| *Caiman* | ROM (no #) | 6.1 | 12.1 | 1.0 | – | Calculated by ilium length (*Dodson, 1975*) | 13.4 | 10.1–16.8 |
| *Allosaurus* | CMN 38454 | 54.8 | 1468.1 | 776.1 | MOR 693 | *Bates et al. (2009a)* | 1083.7 | 812.8–1354.7 |
| *Albertosaurus* | CMN 11315 | 67.5 | 1952.6 | 910.6 | – | *Mallon et al. (2020)* | 1561.5 | 1171.1–1951.9 |
| Ornithomimdae indet. | TMP 1981.022.0025 | 47.8 | 870.5 | 433.2 | ROM 852 | *Campione & Evans (2020)* | 130.5 | 97.9–163.1 |
| *Falcarius* | CEUM 77189 | 28.1 | 316.1 | 76.2 | UMNH VP 12361 | *Campione & Evans (2020)* | 86.9 | 65.2–108.6 |
| Caenagnathidae indet. | TMP 1979.020.0001 | 25.5[a] | 295.0 | 85.9 | – | *Funston (2020)* | 66.8 | 50.1–83.5 |
| *Saurornitholestes* | UALVP 55700 | 19.2 | 91.9 | 26.4 | TMP 1988.121.0039 | *Campione & Evans (2020)* | 18.7 | 14.0–23.4 |
| *Sinovenator* | IVPP V12583/V12615 | 6.9 | 19.5 | 4.1 | IVPP V12615 | Pers. obs. (P. Currie, 2001) | 1.5 | 1.1–1.8 |
| Troodontidae indet. | UALVP 55804 | 30.3 | 455.0 | 163.1 | CMN 12340 | *Benson et al. (2014)* | 119.5[b] | 89.6–149.3 |
| *Struthio* | UAMZ 7159 | 62.7 | 764.7 | 368.4 | – | *Olson & Turvey (2013)* | 115.0[c] | 86.3–143.8 |
| *Dromaius* | UAMZ B-FIC2014.260 | 46.4 | 445.5 | 219.4 | – | *Olson & Turvey (2013)* | 36.9[c] | 27.7–46.1 |
| *Gallus* | RM 8355 | 10.0 | 26.9 | 5.3 | – | *Allen et al. (2013)* | 2.6 | 2.0–3.3 |

**Notes.**
[a] underestimated because the anterior portion of ilium is broken.
[b] body mass doubled based on hip size (subjective estimate).
[c] species mean chosen to match sex of specimen (ilium).

**Table 8  Summary of Phylogenetic Generalized Least Squares (PGLS) regressions.** Study taxa are grouped according to the dependent variable (y) tested against body mass (x), both of which were log-transformed before analysis. Because the linear regression equations can be expressed as $y = bx^k$, the slope (k) can be directly compared to scaling factors, which state that area increases as (body mass)$^{2/3}$ and length increases as (body mass)$^{1/3}$.

| Dependent variable | $n$ | $R^2$ | $R^2_{adj}$ | k | b | $p$-value |
|---|---|---|---|---|---|---|
| All hip muscles (cm$^2$) | | | | | | |
| Non-theropod Sauria | 3 | 0.9919 | 0.9838 | 1.3166 | −0.3549 | 0.0574 |
| Non-maniraptoran Theropoda | 3 | 0.9541 | 0.9083 | 0.3120 | 2.2460 | 0.1374 |
| Non-avian Maniraptora | 5 | 0.9939 | 0.9919 | 0.7200 | 1.1132 | 0.0002 |
| Aves | 3 | 0.9673 | 0.9345 | 0.8907 | 1.0964 | 0.1158 |
| Non-avian Theropoda | 8 | 0.9378 | 0.9275 | 0.6739 | 1.1650 | $7.6860 \times 10^{-5}$ |
| All bipeds | 11 | 0.9294 | 0.9215 | 0.7241 | 1.0187 | $1.7650 \times 10^{-6}$ |
| All taxa | 14 | 0.9165 | 0.9095 | 0.8083 | 0.5138 | $7.9320 \times 10^{-8}$ |
| Major extensors (cm$^2$) | | | | | | |
| Non-theropod Sauria | 3 | 0.9533 | 0.9067 | 1.4593 | −1.7517 | 0.1386 |
| Non-maniraptoran Theropoda | 3 | 0.9951 | 0.9901 | 0.2952 | 2.0020 | 0.0448 |
| Non-avian Maniraptora | 5 | 0.9918 | 0.9891 | 0.8290 | 0.3274 | 0.0003 |
| Aves | 3 | 0.9475 | 0.8950 | 1.1295 | 0.3170 | 0.1471 |
| Non-avian Theropoda | 8 | 0.9022 | 0.8859 | 0.7878 | 0.5231 | 0.0003 |
| All bipeds | 11 | 0.8897 | 0.8775 | 0.8679 | 0.2893 | $1.3310 \times 10^{-5}$ |
| All taxa | 14 | 0.8611 | 0.8495 | 1.0258 | −0.9182 | $1.7270 \times 10^{-6}$ |
| Ilium length (cm) | | | | | | |
| Non-theropod Sauria | 3 | 0.9854 | 0.9709 | 0.3455 | 0.4118 | 0.0770 |
| Non-maniraptoran Theropoda | 3 | 0.7912 | 0.5825 | 0.1262 | 1.3804 | 0.3021 |
| Non-avian Maniraptora | 5 | 0.9903 | 0.9870 | 0.3326 | 0.8077 | 0.0004 |
| Aves | 3 | 0.9684 | 0.9368 | 0.4884 | 0.8168 | 0.1137 |
| Non-avian Theropoda | 8 | 0.9166 | 0.9028 | 0.3108 | 0.8284 | 0.0002 |
| All bipeds | 11 | 0.8753 | 0.8615 | 0.3498 | 0.7148 | $2.3300 \times 10^{-5}$ |
| All taxa | 14 | 0.8779 | 0.8677 | 0.3898 | 0.3871 | $7.9110 \times 10^{-7}$ |

**Notes.**

b, intercept; k, slope; $n$, sample size; $R^2$, multiple R-squared value; $R^2_{adj}$, adjusted R-squared value.

Phylogenetically corrected residual outputs for non-avian theropods and for bipedal taxa were calculated as a percentage of the fitted (predicted) value to examine how they plot relative to each regression line with respect to body size (Table 9). Essentially, this allows clearer visualization of how far above or below the predicted value each taxon plots for the area of all hip muscles, the area of major extensors, or the size of the hip (Fig. 12). For non-avian Theropoda, these charts show that Ornithomimidae indet. has significantly (>5%) higher than predicted values for all three categories at its estimated body mass (Fig. 12). In contrast, *Saurornitholestes* plots significantly lower than the predicted value in all three aspects, and *Sinovenator* also scores significantly smaller than average for ilium length (Table 9). No other taxa significantly deviate from the non-avian theropod PGLS regression line. For all bipeds, the results are strikingly different (Fig. 12). Several taxa have residuals greater than ±5% the bipedal regression line and most of the non-avian

theropods have residuals that are greater in magnitude and opposite in direction from those in the non-avian theropod regression (Table 9).

## DISCUSSION

Inspection of the pelves of maniraptoran theropods that were previously understudied or unstudied for pelvic myology (Fig. 1) allowed identification of osteological correlates for pelvic soft tissues (Figs. 2–5). Besides documenting some morphological conditions that appear to be unique among Archosauria, these osteological correlates also provided the necessary data for novel reconstructions of pelvic musculature (Fig. 6) for comparison to other theropods and extant relatives (Fig. 7). In turn, the area of each origin was quantified, which permits comparison among taxa for the relative sizes of the areas of attachment of anatomical groups (Fig. 8) and functional groups (Figs. 9–10). This revealed a more complex pattern of evolution than previously appreciated, especially in the broader context of other non-avian theropods and their extant relatives. Whereas non-maniraptoran theropods share similar proportions of pelvic muscle origin areas in both anatomical and functional aspects (Figs. 8–10), non-avian maniraptorans deviate strongly from these proportions and from one another (*Allen et al., 2013*).

Statistical analysis yielded no significant difference between the sizes of origin areas acquired by Corel DRAW! or ImageJ. This validates the use of either program for measuring the area of osteological correlates and eliminates method choice as a potential source of bias. However, quantifying the area of attachment is only one way to measure muscles. Certain measurements (e.g., physiological cross-sectional area, mass, tendon length), or combinations thereof, may be better predictors of muscle strength or force (*Bamman et al., 2000*; *Jones et al., 2008*), although these measurements are generally not feasible in fossilized specimens. There is also debate about the reliability of muscle reconstruction in extinct animals, including the ability to glean relative size or strength from attachment area alone (*McGowan, 1979*). The current study is not exempt from these issues; for example, the flexor cruris group occupies merely 2–3% of pelvic origin areas across extant squamates and crocodylians (Fig. 8A) despite accounting for up to 19% of hind limb musculature volume in lizards (*Russell & Bauer, 2008*). In light of these concerns, it should be clarified that our goal is not to attempt estimation of strength, mass, or any other absolute measurement of muscular capacity. Exploration of potential relationships between the area of attachment of a muscle and other muscular properties exceeds the scope of this study and is deserving of its own investigation. Instead, the proportional areas of muscle origins are simply placed in a comparative framework to examine relative changes and explore how well they align with other methods, such as the proportional lengths of limb bones.

Normalizing data to body mass provided an opportunity to explore the effect of body size on the area of attachment and pelvis size. The sampled taxa span three orders of magnitude of body mass and the results follow known scaling principles (*Biewener, 1989*), which makes this evaluation applicable to a wide range of body sizes. Nevertheless, these results should be interpreted with caution as they are based on a small sample size and do not consider other attributes that affect locomotion, such as posture or gait (*Sellers et al.,*

**Table 9  Phylogenetically corrected residual outputs from select Phylogenetic Generalized Least Squares (PGLS) regressions (Table 8).** Study taxa are grouped according to the dependent variable (y) tested against body mass (x), both of which were log-transformed before analysis. Residuals were calculated as percentages of fitted values to adjust for body mass.

| Dependent variable | Non-avian Theropoda | | | All bipeds | | |
|---|---|---|---|---|---|---|
| | Fitted value | Residual | Percentage of fitted value (%) | Fitted value | Residual | Percentage of fitted value (%) |
| **All hip muscles (cm²)** | | | | | | |
| *Allosaurus* | 3.2104 | 0.0281 | 0.8754 | 3.2162 | −0.0495 | −1.5384 |
| *Albertosaurus* | 3.3173 | 0.0562 | 1.6945 | 3.3311 | 0.0765 | 2.2952 |
| Ornithomimidae indet. | 2.5908 | 0.2015 | **7.7793** | 2.5506 | −0.2244 | **−8.7994** |
| *Falcarius* | 2.4718 | 0.0191 | 0.7732 | 2.4227 | −0.1377 | **−5.6844** |
| Caenagnathidae indet. | 2.3949 | −0.0103 | −0.4319 | 2.3402 | −0.05 | −2.1354 |
| *Saurornitholestes* | 2.0225 | −0.1811 | **−8.9553** | 1.94 | 0.1431 | **7.377** |
| *Sinovenator* | 1.2775 | 0.0149 | 1.1652 | 1.1396 | −0.0443 | −3.891 |
| Troodontidae indet. | 2.5649 | 0.0267 | 1.0397 | 2.5228 | 0.1723 | **6.8289** |
| *Struthio* | | | | 2.5108 | 0.075 | 2.9879 |
| *Dromaius* | | | | 2.1534 | 0.0083 | 0.3858 |
| *Gallus* | | | | 1.3192 | 0.0663 | **5.024** |
| **Major extensors (cm²)** | | | | | | |
| *Allosaurus* | 2.9139 | 0.0269 | 0.9234 | 2.9232 | −0.0333 | −1.1402 |
| *Albertosaurus* | 3.0388 | 0.0688 | 2.264 | 3.0609 | 0.0563 | 1.8405 |
| Ornithomimidae indet. | 2.1897 | 0.2986 | **13.6363** | 2.1254 | −0.3035 | **−14.2809** |
| *Falcarius* | 2.0506 | −0.0712 | −3.4704 | 1.9721 | −0.2135 | **−10.8262** |
| Caenagnathidae indet. | 1.9608 | −0.0532 | −2.7132 | 1.8732 | −0.0898 | −4.7928 |
| *Saurornitholestes* | 1.5254 | −0.2659 | **−17.4295** | 1.3936 | 0.2458 | **17.6412** |
| *Sinovenator* | 0.6546 | 0.0221 | 3.3736 | 0.4342 | −0.1022 | **−23.5423** |
| Troodontidae indet. | 2.1594 | −0.0492 | −2.2805 | 2.0921 | 0.324 | **15.4869** |
| *Struthio* | | | | 2.0777 | 0.0607 | 2.9195 |
| *Dromaius* | | | | 1.6493 | 0.0472 | 2.863 |
| *Gallus* | | | | 0.6495 | 0.0391 | **6.0274** |
| **Ilium length (cm)** | | | | | | |
| *Allosaurus* | 1.7718 | 0.0345 | 1.9463 | 1.7765 | −0.0377 | −2.1216 |
| *Albertosaurus* | 1.8211 | 0.0345 | 1.895 | 1.832 | −0.0384 | −2.0965 |
| Ornithomimidae indet. | 1.486 | 0.0767 | **5.1638** | 1.4549 | −0.1582 | **−10.8767** |
| *Falcarius* | 1.4312 | −0.0079 | −0.5517 | 1.3931 | −0.0638 | −4.5786 |
| Caenagnathidae indet. | 1.3957 | −0.0552 | −3.9533 | 1.3532 | −0.0393 | −2.9077 |
| *Saurornitholestes* | 1.224 | −0.094 | **−7.6767** | 1.1599 | 0.1012 | **8.7271** |
| *Sinovenator* | 0.8803 | −0.0527 | **−5.9886** | 0.7731 | −0.0162 | −2.0964 |
| Troodontidae indet. | 1.4741 | −0.0031 | −0.2091 | 1.4414 | 0.1189 | **8.2519** |
| *Struthio* | | | | 1.4357 | 0.0114 | 0.7943 |
| *Dromaius* | | | | 1.263 | −0.0562 | −4.4489 |
| *Gallus* | | | | 0.8599 | 0.0276 | 3.2115 |

**Note**
A residual exceeding ±5% of the fitted value was considered significant and is bolded.

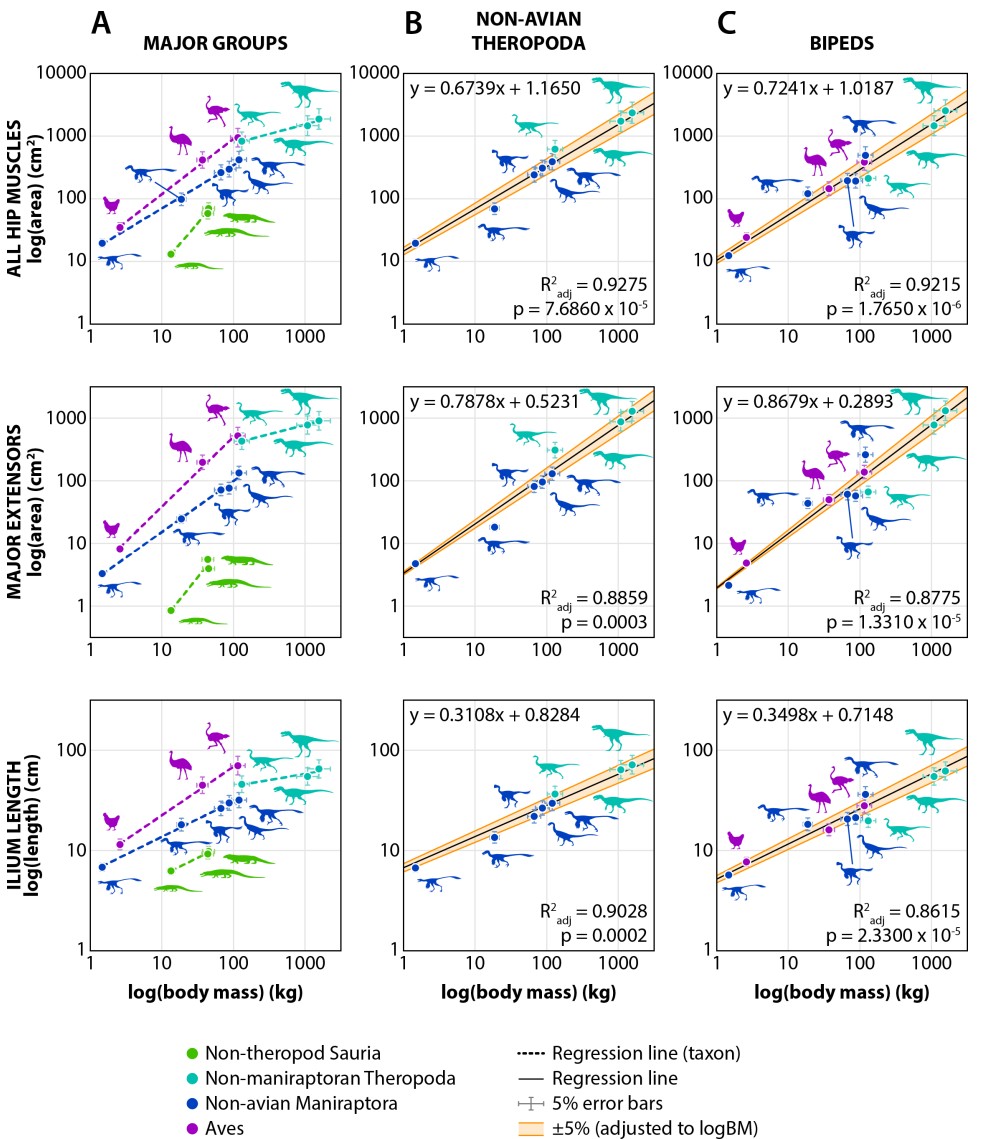

**Figure 12 Phylogenetic Generalized Least Squares (PGLS) regressions on area of attachment and ilium length normalized to body mass (all log-transformed).** (A) Individual regressions on each major group with colour-coded regression lines (dashed). (B) Regressions on non-avian theropods. (C) Regressions on bipedal taxa. Bivariate comparisons are grouped by rows according to the dependent variable: area of attachment of all hip muscles, area of attachment of major extensors, and length of the ilium.

*2017*; *Bishop et al., 2018a*). Examining these effects is beyond the scope of this study and consequently limits explanatory power. Despite these shortcomings, the area of all pelvic muscle origins and the length of the ilium in non-avian theropods closely follow scaling principles (Table 8) (*Biewener, 1989*; *Hutchinson & Garcia, 2002*). Although the regressions for all bipeds tend to follow similar patterns, the phylogenetically corrected residuals casts doubt on how well it represents the data. Many residual values exhibit a higher magnitude and opposite direction, and a greater number of values are outside the ±5% range,

compared to the residuals of the non-avian theropod regression (Table 9). This contrast may stem from the radically different anatomical and locomotory adaptations between non-avian theropods and extant birds, such as postural differences that affect the entire terrestrial locomotory system (*Hutchinson & Gatesy, 2000*; *Bishop et al., 2018b*). Because of these reasons, we henceforth focus on the non-avian theropod PGLS regression results. The regression line for the area of attachment for major extensors is less straightforward, but seems to indicate that major extensors scale with body size at a different rate than the total area of all pelvic musculature. The slope for major extensors in bipeds ($k = 0.7878$) suggests that these muscles increase at a faster rate than all muscles ($k \approx 0.67$), which is closer to isometric growth but still negatively allometric. This is consistent with previous studies that suggest large theropods, despite some having multiple adaptations for fast or efficient running, had relatively poorer running abilities due to large body size (*Hutchinson & Garcia, 2002*; *Dececchi et al., 2020*). Large body mass generally requires larger muscles to overcome inertia, but our analysis cannot identify how this affects running ability. Therefore, even though the large theropods *Allosaurus* and *Albertosaurus* each have areas of hip muscles and of major extensors within 5% of the value predicted from the non-avian theropod regression (Fig. 12) (Table 9), negative allometry may have inhibited attaining the musculature required for rapid locomotion at large body sizes in these taxa (*Hutchinson & Garcia, 2002*). Despite this caveat, relative comparisons among the taxa studied can still be investigated.

Examining the distribution of the non-avian theropod residuals (Table 9) essentially shows whether each taxon exhibits areas of muscle origins or hip size that are larger or smaller than predicted for its estimated body mass (Fig. 12). The only taxa with residuals significantly above or below average are Ornithomimidae indet., *Saurornitholestes*, and *Sinovenator* (Table 9). Perhaps unsurprisingly, the "ostrich-mimic" has higher-than-predicted values in all three categories (Fig. 12), consistent with the other adaptations for running seen in ornithomimids (*Russell, 1972*; *Paul, 1998*; *Carrano, 1999*). The residuals in all categories for *Saurornitholestes* are notably smaller, which is also consistent with its known reduction in cursorial adaptations (*Carrano, 1999*; *Persons & Currie, 2012*; *Persons & Currie, 2016*). *Sinovenator* was significantly smaller in ilium length (Table 9) but the derived troodontid was nearly on the regression line, which reinforces the secondary expansion of the hip in Troodontidae; scaling these measurements to body mass indicates that they are likely not simply an artefact of allometry. The remaining non-avian theropods have residuals within the ±5% range, indicating that these metrics are within expectations for each taxon based on its body mass after phylogenetic correction (Fig. 12).

## Maniraptoran pelvic myology indicates complex evolutionary patterns

Compared to hips of earlier-branching theropods like tyrannosaurids and ornithomimids (Fig. 7) (*Russell, 1972*; *Paul, 1998*; *Carrano, 2000*; *Hutchinson, 2001a*; *Carrano & Hutchinson, 2002*; *Hutchinson et al., 2005*; *Macdonald & Currie, 2019*), *Falcarius* is much smaller in the postacetabulum and brevis fossa (Figs. 2A–2B). These smaller areas available for muscles (Fig. 6A) are mirrored by smaller hip extensors with origins in these regions (Fig. 9A). Among these extensors is the M. caudofemoralis brevis, which

has implications for the evolution of locomotor modules in theropods (*Gatesy & Dial, 1996*). The caudofemoral muscles comprise an important hind limb extensor complex in earlier theropod groups (*Gatesy, 1990*; *Hutchinson & Gatesy, 2000*; *Persons & Currie, 2011a*; *Persons & Currie, 2017*), which can be seen in the relatively large proportion of pelvic musculature taken up by the origin of M. caudofemoralis brevis as a proxy for this complex in *Allosaurus* and *Albertosaurus* (Fig. 8). Even ornithomimids retain a comparable origin of M. caudofemoralis brevis (Fig. 8) despite being the earliest group to show decreases in the number of caudal vertebrae and in tail length and, by extension, the caudal locomotor module (*Gatesy, 1990*; *Gatesy & Dial, 1996*). *Falcarius*, with an estimated 30–35 caudal vertebrae (*Zanno, 2010a*), has a similar vertebral count and presumably similar tail length to an ornithomimid, but shows a sharp reduction in the origin of M. caudofemoralis brevis (Fig. 8). Rather than a stepwise change, this transition appears rather abrupt, at least for the M. caudofemoralis brevis. It suggests less integration between the caudal and pelvic locomotor modules than in earlier theropod lineages, although not fully decoupled from one another (*Gatesy & Dial, 1996*).

Caenagnathids resemble *Falcarius* in having grossly similar osteological and myological features (Figs. 3, 6B, 8) (*Zanno, 2010a*). The reduced postacetabulum and brevis fossa have the same implications for locomotor modules, which reinforces a strong reduction in the tail and its associated musculature within Maniraptora, perhaps even among its earliest members (Fig. 8) (*Allen et al., 2013*). Additionally, caenagnathids exhibit a unique osteological correlate for the M. puboischiofemoralis externus 2 on the posterior sides of the pubes, displaced laterally from the condition in other non-avian theropods (including other oviraptorosaurs) with no evidence of muscle attachment on the pubic apron (Fig. 6B) (*Rhodes, Funston & Currie, 2020*).

*Sinovenator* has a small pelvis relative to its body size (Fig. 12) and a transversely broad pubic apron, both of which are representative of other early troodontids (*Russell & Dong, 1993*; *Currie & Dong, 2001*; *Xu et al., 2002*; *Xu et al., 2017*; *Shen et al., 2017a*). This restricts the area of attachment for most pelvic musculature while increasing the relative area of the Mm. puboischiofemorales externi 1–2 on the pubic apron (Table 4). Surprisingly, this appears as a dramatic change when comparing all pelvic musculature (Fig. 8A) but is almost indistinguishable from bracketing taxa when individual muscles are compared within anatomical groups (Fig. 8B). Derived troodontids underwent a reversal in morphology that is echoed in the myology (Figs. 5–6). Expansion of the postacetabulum, brevis fossa, and ischium (including the obturator process) allowed for respectively enlarged origins of the M. iliofibularis and M. flexor tibialis externus, M. caudofemoralis brevis, and Mm. adductores femorum 1–2 and M. puboischiofemoralis externus 3 (Fig. 6). Many of these muscles are involved in hind limb extension (Fig. 9A), a secondary increase that would have gone unnoticed if *Sinovenator* had been excluded (Fig. 9B).

In derived troodontids, the laterally expanded origin of M. puboischiofemoralis externus 2 is absolutely larger (Table 3) despite becoming proportionately smaller (Table 4). This is likely due, in part, to the relative increase in hip extensors (Figs. 6D, 9). Lateral expansion or migration of the origin of M. puboischiofemoralis externus 2 in troodontids is shared with caenagnathids and birds. However, all three cases appear to arise via

convergence, having developed after each clade diverged from other theropods. This origin in caenagnathids does not seem to be shared with other oviraptorosaurs and thus arose independently (*Hutchinson, 2001a*; *Rhodes, Funston & Currie, 2020*), migrating laterally and leaving no evidence of attachment on the pubic apron (Figs. 3, 6B). The origin of M. puboischiofemoralis externus 2 in troodontids began expanding but remained on the pubic apron in *Sinovenator*, widening further in derived troodontids (Figs. 4, 6C–6D). The evolutionary pathway in early birds remains somewhat unclear, although it eventually moved entirely off the bony pelvis and onto the puboischiadic membrane in birds as the M. obturatorius medialis (*Hutchinson, 2001a*; *Hutchinson, 2002*). Evidently, on three separate occasions, this muscle independently underwent lateral movement in theropods, which may be related to postural or functional changes (*Hutchinson & Gatesy, 2000*; *Bishop et al., 2018b*).

Beyond non-avian Maniraptora, modern birds exhibit notable variability in the number of pelvic muscles and their areas of attachment (Tables 3–4; Figs. 8–9). Similarly, there is substantial diversity in the pelvic morphology and myology of extant avians (Figs. 7EE–7JJ). The contrast between this observed complexity and previously established hypotheses of prolonged, stepwise change over evolutionary time (*Hutchinson, 2002*; *Hutchinson & Allen, 2009*; *Bishop et al., 2018a*) may suggest that the latter perspective is an oversimplification to some extent. Others have noted elevated rates of morphological evolution in birds and their stem lineage (*Allen et al., 2013*; *Dececchi & Larsson, 2013*; *Brusatte et al., 2014*), likely accounting for part of this complexity. Interestingly, *Brusatte et al. (2014)* found that significantly high rates of morphological change appeared near the origin of Maniraptora and were sustained into early birds. This is generally consistent with our results, which juxtapose gross morphological and myological similarity among the pelves of non-maniraptoran theropods with greater complexity in maniraptorans, including avians (Figs. 6–9). Although the overall pattern of incremental changes seems to hold true, our results emphasize that the rate of change was not as straightforward.

### Inferring running ability from musculature

Although there is a complex relationship between the area of attachment and the cross-sectional area, strength, or moment arm of a muscle (*McGowan, 1979*; *Bamman et al., 2000*; *Jones et al., 2008*), hind limb extensors are fundamental to cursoriality. Broadly defined, cursoriality is a spectrum of locomotor ability, recognizable by a suite of complementary morphological features (*Carrano, 1999*). More cursorial animals are better adapted for fast or efficient locomotion, although specific cursorial styles (e.g., sprinting vs. endurance running) cannot usually be elucidated without other means of observation (*Carrano, 1999*). Cursorial theropods typically have reduced arms, slender limbs, long distal leg elements relative to the femur, hinge-like joints, tightly appressed or fused metatarsi, symmetrical feet, lateral pedal digit reduction or loss, and elongate middle toes (*Lull, 1904*; *Coombs, 1978*; *De Bakker et al., 2013*; *Lovegrove & Mowoe, 2014*; *Persons & Currie, 2016*; *Dececchi et al., 2020*). Additionally, cursorial animals tend to have proximally positioned insertions of locomotory muscles, specifically those involved in hind limb extension (*Gatesy, 1990*; *Gatesy & Dial, 1996*; *Carrano, 1999*; *Hutchinson & Gatesy, 2000*; *Hutchinson, 2006*; *Hutchinson &*

*Allen, 2009*). Although insertion points were not considered in this analysis, the areas of origin of hind limb extensors still offer information. These include muscles that inserted on the femur to directly retract the leg, or knee flexors that inserted on the shin to secondarily cause the same motion. The only muscles that function in hip extension across all taxa are the triceps femoris extensors (squamate M. iliotibialis, crocodylian Mm. iliotibiales 2–3, and avian M. iliotibialis lateralis), M. iliofibularis, flexor cruris group, and short head of the caudofemoral complex (Table 5). Presenting traditional cursorial categories sensu *Carrano (1999)* in parallel with the origin areas of hip extensors show that these two aspects are remarkably well correlated (Fig. 13).

Individual extensors are not 100% reliable proxies for cursoriality; although the M. iliofibularis reflects relative cursorial categories, the M. caudofemoralis brevis suggests that caenagnathids are less cursorial than dromaeosaurids (Fig. 13), which conflicts with the general results of other studies noting decreased running ability in the latter (*Carrano, 1999*; *Fowler et al., 2011*; *Persons & Currie, 2012*; *Persons & Currie, 2016*). It is possible that this is due to differences in cursorial styles, such as sprinting vs. endurance running (*Dececchi et al., 2020*), but this cannot currently be differentiated by the analysis here. On the other hand, the combination of the M. iliofibularis and M. caudofemoralis brevis better reflects cursoriality across taxa, showing a similar pattern to the inclusion of all hip extensors (Fig. 13). Jenks Natural Breaks optimization performed on the proportion of major extensors recovers five classes with a high goodness of variance fit ($k = 5$, GVF $= 0.9843$) (Fig. 13). The same analysis conducted with four classes to reflect traditional cursorial categories (*Carrano, 1999*) reveals a slightly lower fit ($k = 4$, GVF $= 0.9752$), although relatively small sample size is probably why both values are high. Comparison of these two classifications as models using the small-sample corrected AIC (*Hurvich & Tsai, 1989*) recovered the five-class analysis (AIC$_C = -53.95$) as having a better fit than the four-class test (AIC$_C = -51.89$). Although the five-class model only scored marginally better ($\Delta$AIC$_C = 2.06$), it is preferred over the four-class model for our data based on these scores (*Burnham & Anderson, 2002*; *Burnham & Anderson, 2004*). The only difference is that the five-class analysis (Fig. 13) separates derived Troodontidae indet. from Caenagnathidae indet. and *Saurornitholestes*, whereas the four-class analysis recovers all three of these taxa in the same category (second-highest class overall). Squamates and crocodylians are unsurprisingly recovered as the poorest cursors of all taxa, whereas non-maniraptoran theropods all score highly based on pelvic muscle origin areas (Fig. 13). Although expected for tyrannosaurids and ornithomimids (*Russell, 1972*; *Paul, 1998*; *Carrano, 1999*; *Persons & Currie, 2016*), it was unanticipated for allosaurids (*Bates, Benson & Falkingham, 2012*). Non-avian maniraptorans vary, though *Falcarius* and *Sinovenator* consistently show lower cursoriality than the caenagnathid, *Saurornitholestes*, and the derived troodontid (Fig. 13). The caenagnathid only slightly exceeds *Saurornitholestes*, which quantitatively supports the hypothesis that caenagnathids are less cursorial than previously thought (*Rhodes, Funston & Currie, 2020*). This result may have previously been masked by categorizing the degree of cursoriality into distinct bins, an issue noted by *Carrano (1999)* that is applicable to any spectrum. The cursoriality of *Saurornitholestes* and derived troodontids match predictions, and inclusion of *Sinovenator* suggests that high cursoriality was gained

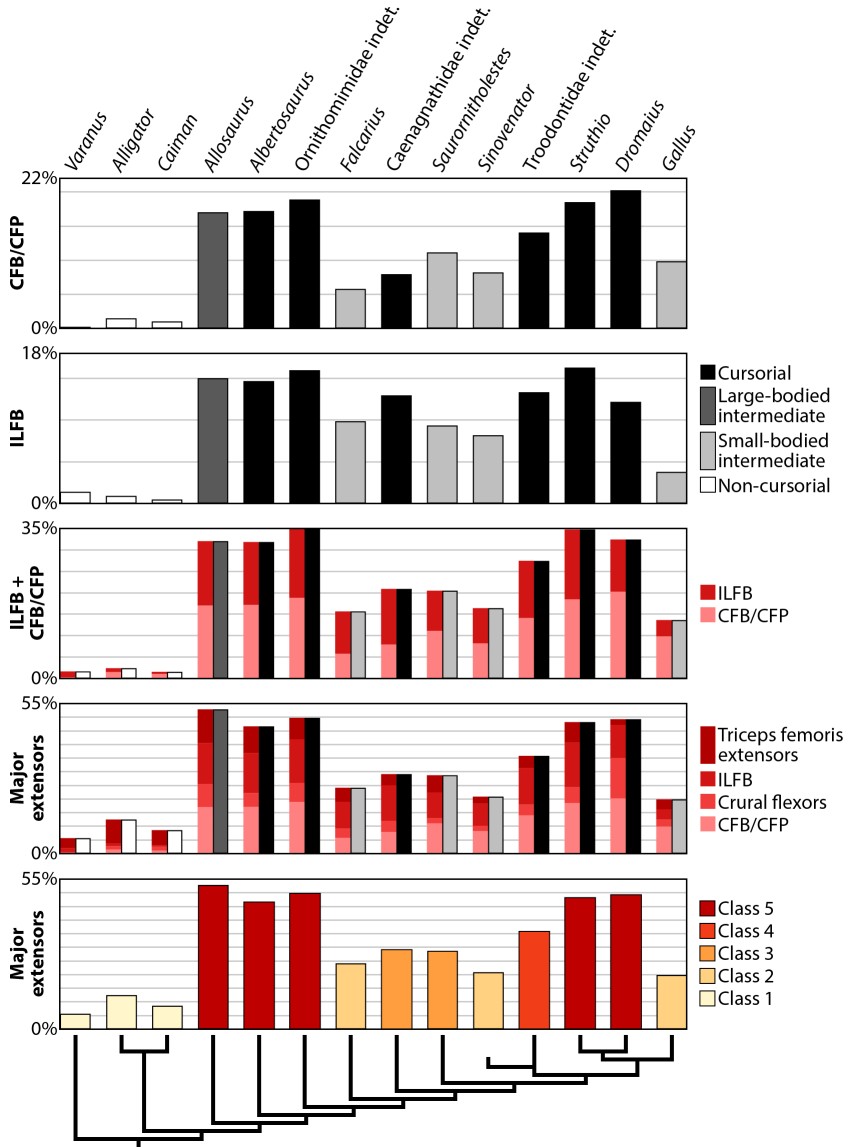

**Figure 13 Area of attachment of major extensors.** Muscles and groups plotted as a proportion of the total area of all pelvic muscle origins, and grey lines represent 5% increments. Cursoriality categories adapted from *Carrano (1999)*. Classes based on Jenks Natural Break optimization of the major extensors for five groups. See Table 2 for muscle abbreviations.

secondarily in Troodontidae (Fig. 13). Among birds, the contrast between the ostrich and the emu as known high-speed runners that exhibit several cursorial adaptations, and the chicken as a relatively poorer runner, corresponds well with their relative major extensor muscle proportions (Fig. 13) (*Patak & Baldwin, 1998*; *Smith et al., 2006*; *Smith et al., 2007*; *Abourachid & Renous, 2008*; *Lamas, Main & Hutchinson, 2014*; *Hutchinson et al., 2015*).

Overall, the results of this study show that the proportion of certain extensor origins is a quantifiable metric that can be used to reasonably infer cursoriality. That being said, cursoriality is multifaceted, reliant on multiple complementary morphological features

as supporting lines of evidence (*Coombs, 1978*; *Carrano, 1999*; *Dececchi et al., 2020*). This point is underscored by comparing pelvic musculature with other morphological features associated with cursoriality, which can be averaged for a more comprehensive view on cursorial ability (Fig. 14). Squamates and crocodylians consistently score as the least cursorial taxa in all metrics (Fig. 14B). Within non-maniraptoran Theropoda, *Allosaurus* superficially appears highly cursorial because 48% of all pelvic muscle origins are occupied by major hip extensors (Table 4; Fig. 13). However, its intermediate proportions of distal limb elements (*Carrano, 1999*), non-arctometatarsalian foot (*Holtz, 1994*), and three weight-bearing digits with a mostly symmetrical foot (*Gilmore, 1920*) all indicate more moderate cursoriality (Figs. 14B–14C), which better corresponds to previous results on allosaurid locomotion (*Bates, Benson & Falkingham, 2012*). Moreover, this may show that theropods attained the musculature necessary for high cursoriality before the acquisition of osteological adaptations like long distal limb proportions, modified metatarsi, or other features (Fig. 14B). However, assessing more basal theropods would help to test this hypothesis. In contrast, both tyrannosaurids and ornithomimids—having large hip extensors, long distal limb bones, and arctometatarsalian feet that are also more symmetrical (*Russell, 1972*; *Holtz, 1994*; *Paul, 1998*; *Carrano, 1999*; *Snively & Russell, 2001*; *Snively & Russell, 2003*; *Henderson & Snively, 2004*; *Persons & Currie, 2016*)—score highly in nearly all aspects (Figs. 14B–14C). Although adult tyrannosaurids may have been less adept at running (*Hutchinson & Garcia, 2002*; *Hutchinson, 2004b*; *Hutchinson et al., 2005*; *Sellers et al., 2017*), these conditions present in a juvenile are predictive of rapid locomotion (*Paul, 1998*; *Persons & Currie, 2011a*; *Dececchi et al., 2020*). Non-avian maniraptorans are variable, marked by a substantial reduction in cursorial features already present in *Falcarius* (Figs. 14B–14C) (*Zanno, 2010a*). Reconstruction of pelvic musculature suggests that caenagnathids are less cursorial than previously predicted (*Rhodes, Funston & Currie, 2020*), although they still rank as relatively high in light of other features (Figs. 14B–14C). *Saurornitholestes* and *Sinovenator* share similar degrees of cursoriality (Fig. 14C). Compared to *Sinovenator*, derived troodontids show expansion of major hip extensors (Table 4; Fig. 13), elongated distal limb segments (*Carrano, 1999*), and development of a subarctometatarsus into a full arctometatarsus (*Holtz, 1994*; *White, 2009*) that corroborate them as adept runners (Figs. 14B–14C). Furthermore, it supports previous observations of divergent evolutionary pathways between eudromaeosaurians and troodontids, and emphasizes that derived troodontids are intermediate in cursoriality between eudromaeosaurians (and all other maniraptorans, in fact) and highly cursorial theropods including tyrannosaurids, ornithomimids, and palaeognaths (Fig. 14C) (*Fowler et al., 2011*). The area of major hip extensors in each bird is a fair indicator of overall cursoriality based on the taxa included here, separating the palaeognaths from the chicken (Figs. 13–14).

Jenks Natural Breaks optimization distinguishes five classes ($k = 5$, GVF = 0.9957) that better reflect cursorial categories based on consideration of multiple adaptations (Fig. 14C). All of the quadrupeds (*Varanus*, *Alligator*, and *Caiman*) comprise the lowest category. The class above includes *Falcarius*, *Saurornitholestes*, and *Sinovenator*, which are inferred as the least cursorial bipeds in this study. *Allosaurus* and *Gallus* are recovered in the middle

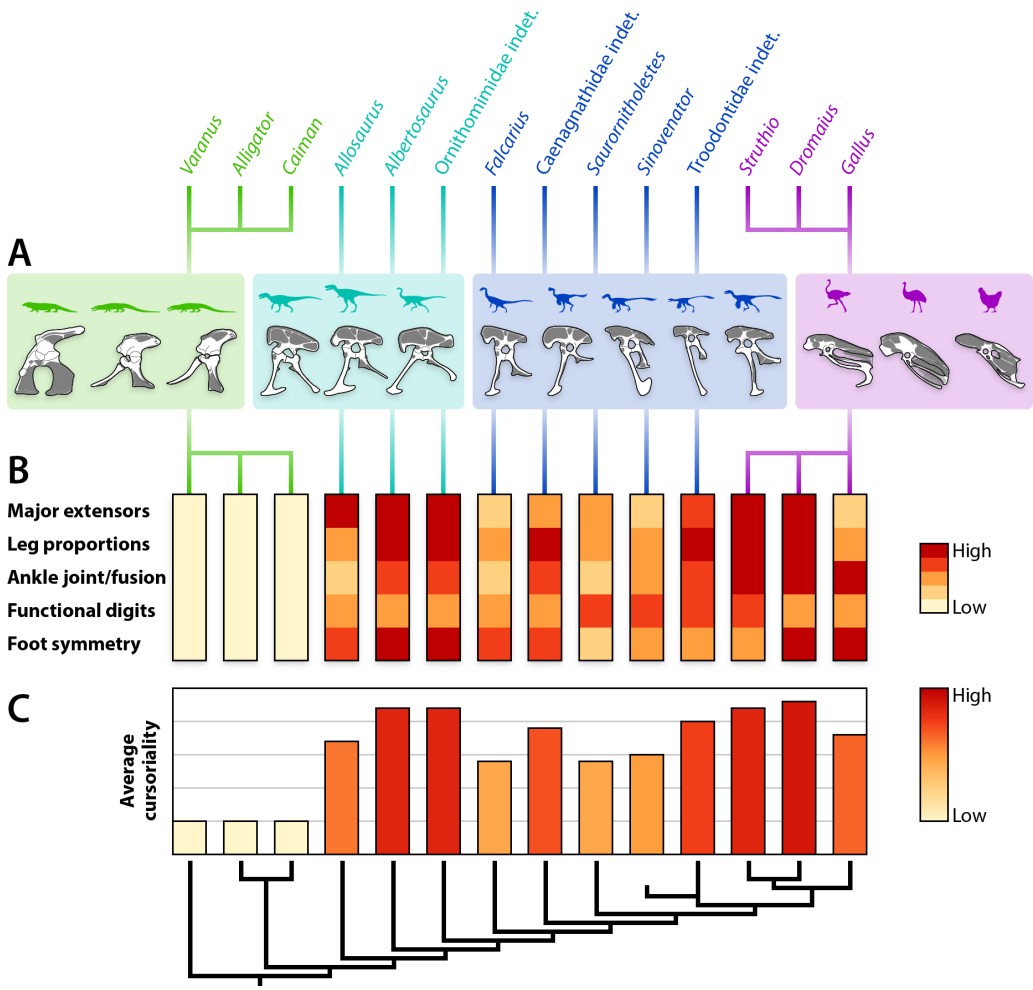

**Figure 14** **Pelvic musculature and other putative correlates of cursoriality.** (A) Pelvic muscle reconstructions in left lateral view grouped to show non-avian maniraptorans among other theropods and extant relatives. (B) Heat map comparing the area of attachment of major extensors (from Jenks Natural Breaks optimization in Fig. 13), proportional length of distal hind limb elements (sensu *Carrano, 1999*), ankle joint morphology/degree of metatarsal fusion (hinge-like > ball-and-socket; tarsometatarsus > arctometatarsus > subarctometatarsus > unspecialized), number of functional weight-bearing digits (fewer = higher), and foot symmetry. (C) Bar chart depicting a spectrum of inferred cursoriality averaged from the heat map above.

category, whereas the caenagnathid and derived troodontid share the second-highest tier. The highest class, representing the taxa best adapted for cursoriality based on this analysis, includes the tyrannosaurid, ornithomimid, ostrich, and emu (Fig. 14C). However, categorizing the degree of cursoriality into four classes also had strong support ($k = 4$, GVF $= 0.9885$). In fact, although the five-class model attains a better fit ($\text{AIC}_C = -2.31$) than the four-class analysis ($\text{AIC}_C = -1.54$), both have substantial support ($\Delta\text{AIC}_C = 0.76$). The only difference between these two options is that the middle and second-highest tier (classes 3 and 4) of the five-class model, containing *Allosaurus*, *Gallus*, Caenagnathidae indet., and derived Troodontidae indet., collapse into a single bin ranked second-highest (class 3)

in the four-class model. Although Jenks Natural Breaks optimization and $AIC_C$ may not appear to clearly distinguish cursoriality, these methods merit further consideration on a larger, more comprehensive dataset.

The results of this study highlight the complex nature of cursoriality (*Dececchi et al., 2020*). Within non-avian Theropoda, the allosaurid, ornithomimid, and tyrannosaurid representatives retain conservative pelvic morphology and myology (Fig. 14A), but their cursoriality differs when accounting for other adaptations (Figs. 14B–14C). Additionally, the reconstruction of ornithomimid pelvic musculature (Fig. 7) offers an updated view of the one provided by *Russell (1972)*, which has not been explicitly shown before despite inclusion of this taxon in volumetric reconstructions (*Bates et al., 2009b*) and musculoskeletal modelling (*Bates & Schachner, 2011*). Maniraptorans tend to exhibit wide variation in both osteology and myology (Fig. 14A). The sharp reduction of caudal musculature in early maniraptorans appears to be a shared state; not only is the origin of M. caudofemoralis brevis reduced in *Falcarius* and the caenagnathid (Figs. 2–3, 6A–6B, 8), the insertion point (fourth trochanter) is also shallow or absent in both of these taxa and more plesiomorphic members of their clade (*Zanno, 2010a*; *Rhodes, Funston & Currie, 2020*). This pronounced reduction contrasts strongly with ornithomimids (*Gatesy, 1990*; *Hutchinson, 2001b*), which suggests that reduction of the tail and its muscles began conservatively at or near the base of Maniraptora rather than sometime within (*Allen et al., 2013*). Although the initial stages of caudal decoupling are apparent as early in coelurosaur evolution as Ornithomimidae (Fig. 1) (*Gatesy & Dial, 1996*), a similarly punctuated step in caudal decoupling probably occurred at the base of Maniraptora in tandem with tail reduction. Further support for partial caudal decoupling comes from caenagnathids, which share similar pelvic morphology and similar relative area of caudal musculature with *Falcarius* (Table 4; Fig. 8) but have a much higher level of cursoriality achieved through enlargement of pelvic muscles—not tail muscles—along with other morphological adaptations (Figs. 13–14).

## Moving forward

Further investigation of other hind limb musculature would allow a more thorough examination of locomotory adaptations. Of the caudofemoral muscles, only the M. caudofemoralis brevis has an osteological correlate on the pelvis for consideration here, but the M. caudofemoralis longus comprises the bulk of this complex (*Gatesy, 1990*; *Persons & Currie, 2011a*; *Hutchinson et al., 2011*). Exploring both parts of the Mm. caudofemorales would provide additional insight into the contribution of caudal musculature to cursoriality and the decoupling of pelvic and tail musculature (*Gatesy & Dial, 1996*). Including other hind limb musculature of the thigh and lower leg would also allow whole-limb reconstruction that would better complement future analyses. Even with these caveats, the results here are largely consistent with previous studies and show that the combined origin areas of major hip extensors are a reasonable proxy for cursoriality, especially in the context of other metrics (Fig. 14). Further study could evaluate potential relationships between origin areas and known muscles properties such as mass, physiological cross-sectional area, maximal isometric force, and others. On a similar note, selection of specimens with

known body masses or associated stylopod elements to estimate body size would refine this analysis and offer better comparisons among taxa. This could be done for multiple specimens of varying body size, within a family or within a species, to better assess potential effects of ontogeny and allometry. Future work could also expand the dataset, including examination of a greater diversity of birds, to better assess potential locomotory trends. A broader dataset would permit a more in-depth analysis of the relationship between major extensors and total hip musculature, which could also incorporate a time-calibrated phylogeny to address questions related to estimated divergence times or ancestral state reconstructions. Furthermore, the methods used here could be applied to other parts of the skeleton to address other myological questions, like how the area of attachment for cranial musculature relates to bite force and feeding strategies, or integrating shoulder and forelimb muscle attachment sites with pelvic and hind limb ones to analyze patterns in quadrupedal locomotion. The results from our relatively small sample merit additional, more robust investigation into the potential relationships between the area of attachment and cursoriality. Although beyond the scope of this study, the areas of attachment of major hip extensors could be integrated into a more comprehensive analysis with other putative correlates of cursoriality, similar to those done for theropod herbivory and tail weaponry (*Zanno & Makovicky, 2011*; *Arbour & Zanno, 2018*; *Arbour & Zanno, 2020*).

## CONCLUSIONS

Examination of the pelvis for osteological correlates of locomotory musculature in the maniraptorans *Falcarius*, Caenagnathidae indet., *Sinovenator*, and derived Troodontidae indet. documented morphological features and highlighted unique, previously unidentified conditions in these groups. In turn, this allowed for novel reconstructions of pelvic myology that could be quantified to compare within and among taxa. Soft tissue inferences provide an additional perspective on locomotory adaptations that often, but do not always corroborate osteological inferences. In *Falcarius*, soft tissue inferences generally match the osteological signals all indicating reduced cursoriality. Caenagnathids, though not as cursorial as previously thought based on pelvic muscles alone, still seem to be competent runners given multiple lines of evidence. *Sinovenator* creates a baseline for the evolution of pelvic musculature in Troodontidae, which supports previous studies noting divergent evolutionary trajectories between this family and eudromaeosaurians (*Fowler et al., 2011*). Overall, our soft tissue inferences are reasonably consistent with previous results indicating a stepwise accumulation of avian-like traits (*Hutchinson & Allen, 2009*). However, rather than an increased rate of anatomical change occurring sometime within Maniraptora (*Allen et al., 2013*), our results suggest that it began with a somewhat punctuated step at the base of this clade. Furthermore, this underscores the increased rates of morphological change seen in Maniraptora, suggesting that the assembly of avian features may have been steady but was not always slow.

Focusing on the origins of pelvic muscles provided a sufficient basis to explore the primary musculature controlling hind limb function, but would be complemented by examination of other caudal and leg muscles in this manner to flesh out a better

understanding of locomotion. Nonetheless, calculating the area of attachment for musculature in a comparative framework offers a quantifiable metric to infer evolutionary patterns and locomotory adaptations. The areas of attachment of major extensors across the taxa examined here are collectively a reasonable proxy for running ability. The hip, as the junction between the hind limb and tail, also provides insight into the integration between locomotor modules (*Gatesy & Dial, 1996*). Based on our results, the decoupling of caudal and pelvic locomotor modules seems to follow the same pattern as pelvic morphology and myology, which underwent a substantial advancement early within Maniraptora. That being said, the caudal module remained at least partially integrated with the pelvis and hind limb throughout non-avian Theropoda, partly contributing to the secondary evolution of high cursoriality in troodontids. Future steps could integrate myological data with osteological data and other methods into a more comprehensive analysis of cursoriality as a spectrum of locomotor ability.

**Institutional abbreviations**

| | |
|---|---|
| **AMNH** | American Museum of Natural History, New York City, New York, USA |
| **BYUVP** | Brigham Young University Museum of Paleontology, Provo, Utah, USA |
| **CEUM** | College of Eastern Utah Prehistoric Museum, Price, Utah, USA |
| **CM** | Carnegie Museum of Natural History, Pittsburgh, Pennsylvania, USA |
| **CMN** | Canadian Museum of Nature, Ottawa, Ontario, Canada |
| **DLXH** | Dalian Xinghai Museum, Dalian, Liaoning, China |
| **IVPP** | Institute for Vertebrate Paleontology and Paleoanthropology, Beijing, China |
| **RM** | Redpath Museum, Montréal, Québec, Canada |
| **MOR** | Museum of the Rockies, Bozeman, Montana, USA |
| **ROM** | Royal Ontario Museum, Toronto, Ontario, Canada |
| **TMP** | Royal Tyrrell Museum of Palaeontology, Drumheller, Alberta, Canada |
| **UALVP** | University of Alberta Laboratory for Vertebrate Palaeontology, Edmonton, Alberta, Canada |
| **UAMZ** | University of Alberta Museum of Zoology, Edmonton, Alberta, Canada |
| **UMNH** | Natural History Museum of Utah, Salt Lake City, Utah, USA |

## ACKNOWLEDGEMENTS

For specimen and collections access, we thank R. Esplin and R. Scheetz (BYUVP), K. Corneli and K. Carpenter (CEUM), J. Mallon, K. Shepherd, and M. Currie (CMN), A. Atwater and J. Scannella (MOR), A. Howell (RM), B. Iwama and K. Seymour (ROM), B. Strilisky and D. Brinkman (TMP), C. Coy and H. Gibbins (UALVP), B. Barr (UAMZ), and C. Levitt-Bussain and R. Irmis (UMNH). For specimen acquisition and dissection assistance, MMR thanks A. McIntosh, B. Barr, C. Sullivan, S. Hamilton, and Y.-y. Wang. Additional helpful discussions and constructive feedback came from M. Caldwell, A. Dyer, D. Evans, G. Funston, R. Holmes, E. Koppelhus, T. Miyashita, A. Murray, M. Powers, and S. Persons. Reviewer feedback has been integral to greatly enhancing previous versions of this project.

### Funding

This work was supported by the Dinosaur Research Institute (MMR); Government of Alberta (MMR); Natural Sciences and Engineering Research Council of Canada (NSERC) Canada Graduate Scholarship (MMR) and Discovery Grant (PJC, No. RGPIN-2017-04715); University of Alberta Department of Biological Sciences (MMR); and University of Alberta Graduate Students' Association (MMR). The funders had no role in study design, data collection and analysis, decision to publish, or preparation of the manuscript.

### Grant Disclosures

The following grant information was disclosed by the authors:
The Dinosaur Research Institute (MMR).
Government of Alberta (MMR).
Natural Sciences and Engineering Research Council of Canada (NSERC) Canada Graduate Scholarship (MMR) and Discovery Grant: PJC, No. RGPIN-2017-04715.
University of Alberta Department of Biological Sciences (MMR).
University of Alberta Graduate Students' Association (MMR).

### Competing Interests

The authors declare there are no competing interests.

### Author Contributions

- Matthew M. Rhodes and Donald M. Henderson conceived and designed the experiments, performed the experiments, analyzed the data, prepared figures and/or tables, authored or reviewed drafts of the paper, and approved the final draft.
- Philip J. Currie conceived and designed the experiments, analyzed the data, authored or reviewed drafts of the paper, and approved the final draft.

### Data Availability

Raw data and code are available in the Supplemental Files.

All specimens described in this manuscript are listed in Table 1 with the identification/accession number and brief description. Institutional abbreviations and locations are listed at the end of the Introduction.

### Supplemental Information

Supplemental information for this article can be found online at http://dx.doi.org/10.7717/peerj.10855#supplemental-information.

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
