# Peer review of "Maniraptoran pelvic musculature highlights evolutionary patterns in theropod locomotion on the line to birds"

_PeerJ, doi:10.7717/peerj.10855_

## Round 0.1 · original submission · Major Revisions

I think that this is a strong manuscript and is deserving of being published after paying careful attention to reviewer comments. Reviewer 2 in particular noted a number of changes and additions. First, I agree with reviewer 2 that the fact that muscle area does not tell you much about muscle size or force production needs to be more clearly stated early on in the manuscript. I think that the manuscript often overreaches in many sections on this point. Although definitely outside of the scope of this manuscript, It would be very cool to compare the Coreldraw and imageJ methods in extant taxa with known PCSA values. Such ideas could be added to your 'moving forward' section. The authors should also be careful to consider allometry in the discussion. As the reviewer notes, radically differently sized animals will have different ratios of muscle attachment sites as a result of their size.

Line 47–49: Reword sentence. Grammar is a little confusing.

Line 312: ‘echoing a similar’

Line 511: I wouldn’t say that it represents less integration, since that would require statistical testing. Change this to ‘it suggests less integration’ or something similar.

Figure 1: Why is Sinovenator lower than the other taxa on the cladogram?

Let me know if you have any questions pertaining to either review. I look forward to receiving your corrections.

Best,

Brandon

·

Basic reporting

This is an exceptionally well written analysis and discussion of the evolution of the pelvic and hindlimb muscles in maniraptoran theropods. I have no major issues with this manuscript, and just a few definitional and organizational suggestions/ questions (below).

Page 6, Line 19: “controlled” is an unusual word choice. Influenced perhaps? Skeletal development is also influenced by genomic and other factors that are not direct musculoskeletal forces.

Page 10, Line 108: “leg” - do you mean thigh? Or the entire hindlimb? I would pick a more specific term – in a lot of anatomical literature the leg is the tib/fib, and thigh is the femoral region.

Results – it would be helpful to add subheadings for each taxonomic group (prior to inferred myology)

Table 5 – perhaps add to the caption that function is derived from previous studies and indicate that these were “averaged” for each muscle across the extant taxa. Were there any differences between sprawling animals and erect bipeds for muscle function in the extant literature? How do the numbers along the top correspond to the taxa in Table 1? This was not immediately obvious to me and could perhaps be clarified in the caption.

Figure 4: A, C, and D – is it possible to get clearer high resolution images of these panels? They are out of focus and very difficult to see.

Figure 6: just curious why this isn’t in color when PeerJ doesn’t charge for color? The Hutchinson etc papers usually use additional color labelling to demonstrate homologous origins/insertions across the various taxa. That would enhance the image.

Figure 7: The font size on this one is VERY tiny and hard to read. If it is at all possible to increase the font size for the key on the right side of the graph/chart and the genera along the top, that would be very helpful. Font size in Fig. 10 is also very very small and should be a little bit bigger so that the figure is readable without zooming in.

With PeerJ being an online journal, some of the supplemental figures could be added to the text. These data are valuable for comparison, and SI images often get lost (or forgotten) after a manuscript is published.

Experimental design

No comment beyond what is stated under Basic Reporting

Validity of the findings

No comment beyond what is stated under Basic Reporting

·

Basic reporting

no comment

Experimental design

no comment

Validity of the findings

no comment

Additional comments

Review of “Maniraptoran pelvic musculature highlights evolutionary patterns in theropod locomotion on the line to birds”


Overall I like this manuscript. You looked at a large number of specimens, extant and extinct, and incorporated that vast knowledge into a clear and strongly written paper. I think the work on looking at the location, extent and how these features shift across phylogeny is an important addition to the cannon of muscle reconstructions in non-avian theropods. You purposely sought out clades that have been left out, and specimens that are smaller than typically done by researchers and this is also needed if we are to understand how the theropod body (not just the bones) evolved. That being said, I do have a couple of major reservations that, while not fatal, need to be at least addressed earlier in the text rather than as a single sentence in the discussion.

Major points:
The first is that simply the morphology of muscle attachment site (entheses) has been shown not to correspond to either muscle size or activity level (Zumwalt 2006; Rabey et al. 2014 amongst others for more recent experimental work). Even in the your own work you cite things like the origin for the M. ambiens 2 in Struthio being unique as it is significantly larger than the origin of the M. iliotibialis, yet the later muscles still has a much larger volume than the combined mass of the M. ambiens 1+2. So while the size of the that attachment is a unique identifier and may be something that can have phylogenetic implications, it cannot be used to reconstruct muscle volume, force production or activity level with any degree of certainty. This needs to me more clearly stated in your paper, how the findings you see about the shifts in the relative areas of the different muscle attachments should not be used to directly infer anything about the muscles size or ability to produce force.

Second, the effects of allometry on this data, especially in the purported shifts within the Troodontids, needs to be addressed. As you scale up you have different needs and constraints that have to be recognized and accounted for before any direct comparison can be made. Hutchinson and Garcia 2002 mention this (see their Table 1) discussing how and adult Tyrannosaurus would need an unnaturally high amount of hindlimbs musculature to run as a proportionally scaled up smaller theropods. While they only have 3 estimates you can clearly see the trend of, as you get bigger, you need relatively more muscle to achieve the minimal to run and overcome the fact that force scales with a lower exponent than mass. Thus when you are comparing the values of a 1-1.5 kg Sinovenator to a Troodontidae indet. that is likely 100-150+ kg ( I used the estimated hip height that is found in van der Reest and Currie 2017 of 1.8 m, which would as the authors state make it the largest known troodontid and over twice the size of MOR 748 which is about 50kg according to Benson et al. 2018) you need to expect that there will be different relative proportions of muscle mass as the minimum power required to make each run is not the same. Also there is the challenge of crouching and how that can shift muscle activity level. I will discuss more when I think it could be mentioned in the text , but when comparing small taxa (less than 2 kg) to much larger ones (over 100 kg) the effects of a more crouched running stance may also influence which muscles are accentuated. These two taxa could both be considered highly cursorial but may show different total levels of relative hindlimb muscle mass as well as changes in the emphasis of those muscles without compromising the ability of the smaller one to be a speedy critter. This has implications for one of the more interesting findings of your paper, the bounce back seen in larger troodontids.

I think both of these topics need to be addressed in more detail, even if it is just to discuss how they remain currently unknown and place limitations on this study, to improve this manuscript.


Minor points:

Most of these are minor issues or are places where acknowledgement of the major ones raised could be appropriate. I have listed the line number per the reviewing document I have for ease of use. If that differs from the authors copy I do apologize.


Line 37- I think that Hutchinson and Allen 2009 should be mentioned in this list of references as well.

Line 41- think that Dececchi et al. 2020 should be mentioned in this list of references as well (sorry I don’t mean to sound like I am trying to force my work into yours, but I think it fits here and you already cite this paper later).

Line 42- I suggest you separate flight and wing based locomotion into separate categories, because they are. Flight origins (can also include Pei et al. 2020 in this list if you wish) looks like it happened at different times in theropods and these origins are distinct from the origins of the use of forelimbs to aid in locomotion such as WAIR or flap running. If you do there are a lot of references for these papers such as those by the Dial lab, Burgers and Chiappe 1999 as well as the Talori et al. 2018, 2019 and Dececchi et al 2016.

Line 47- Maybe mention here how each of these methods has a domain of information but has limitations (like you talk about your method does, you want to make sure that the reader understands that no one method is the silver bullet) . Even a trackway, the direct representation of the motion of an organisms, may not be representative of its most common habitual functional aspects such as speed, stride length and maneuverability. It also cannot give you the range of normal daily functions let alone the upper bounds.


Line 51- Are you using these as historical markers only? Perhaps it may help to illustrate how the methods and mindset has changed (or not) in the almost a 100 years since Romer. Do we use different models? How does phylogenetic bracketing shift what we thought we knew? I am not saying to avoid acknowledge where we came from, but that stand alone sentence really adds nothing to the paper unless you add context to it.

Line 54-I do not like the wording of this sentence. Since theropods are bipeds and, up until the origins of flight, terrestrial then all locomotor studies (not just the initial ones) can only focus on the hindlimb regions. I think it would be best to simply state something like (in your own words and writing style of course) “ Since those pioneering studies significant research has been devoted to understanding the patterns of muscle origins and insertion site at the clade level such as …….. “.

Line 70- I believe Hutchinson and Allan 2009 should be included in this citation as it as another follow up that while a review does discuss some newer topics and synthesizes the data across studies to that point.

Line 76- That what represents a series of gradational changes? All theropod anatomy? Please be a little more specific here .

Line 89- Well, perhaps in some there was reduced running ability, whether you mean long distance vs top speed as tools you need for one may not be required for the other, but the microraptorines have extremely long legs. Legginess effects top speed at small sizes more than distal limb segment proportions, suggesting they are among the speediest. Some of these taxa such as Microraptor also have evolved a subarctometatarsalian condition which has been linked to increase running ability. What you are talking about is the Eudromaeosauria (maybe even just the Velociraptorinae given the likely cursoriality of Bambiraptor) seemingly showing less cursorial adaptations than some comparably sized taxa such as troodontids or ornithomimids. But again that is likely more due to them not being long distance pursuit predators than not being relatively speedy. Lions and tigers and bears are not built like the classical runner, but are surprisingly fast in short bursts. Maybe change it to “pursuit ability” as opposed to running ability.

Line 94- Again I hate to do this but Dececchi et al. 2020 should be mentioned here.

Line 178- This is where you need to address the disconnect between origin and insertion area and muscle size head on. Since enthesis morphology does not correlate to activity level, my suggestion is to say something along the lines of '”yes, we understated that this may not be the clearest way to determine relative muscle size, importance or activity level but this method does provides one way to look at how muscle evolution occurred across non-avian theropods. While not ideal, the fact remains that other methods such as examining internal bone geometry is very limited and sampling is not possible for many critical specimens. Thus we are following previous workers in this field in suggesting some signal should be retained from origin area differences, it is available from a large and diverse set of theropods and that it can be incorporated into a more holistic method for ascertain behavioral categories amongst theropods”. You need to acknowledge clearly and up front the limitations of this methods before proceeding on.

Line 220- There were no conflicting reconstructions found using this? All bracketing taxa showed the exact same morphology? If there were conflicts how were they resolved? Did you assign that reconstruction a lower weighting?

Line 236- I feel “ more basal “ sounds nicer than “ earlier diverging” but it’s not a major concern.

Line 334- There is a size difference as well as Sinovenator is much smaller.

Line 348- How are you accounting for allometry? Sinovenator or Jianianhualong are likely less than 2 kg and the derived troodontids are orders of magnitude more massive. So do you factor that in or are you just looking at the relative size vs femur length or SVL measured as a ration without seeing if there is a scaling factor?

Line 370- That’s not true, as you scale relative scores can change as well. You need proportionally more muscle to run faster at larger sizes than smaller. You cannot assume that because a full sized Tyrannosaurus has relatively larger attachment sites and muscle mass(even assuming they faithfully indicate muscle size) that it is a “better runner” than a smaller theropods because the Tyrannosaurus needs much more absolute and relative muscles to overcome the forces needed to start a run.

Line 391- But this isn’t true for the muscle itself . The M. iliotibialis accounts for 61% of the total triceps femoris according to Hutchinson et al. 2015. So while the origin area is larger for the M. ambiens in Struthio it looks like the muscle size is more in line with the proportions in other taxa.

Line 471-Again you can’t make this statement using attachment site area.

Line 475- Well size probably has an influence here, as the minimum power requirements may not scale the same for all muscles. Your non-avians dataset are mostly big 1000kg+, in contrast your maniraptorans top out at about 130kg (Falcarius has a femur of ~403 mm for the largest know specimen per Lautenschlager et al. 2012). While this is likely similar in size to the largest specimens of Ornithomimus, it is much smaller than the other non-avians used. So, how much does that sampling issue factor into the patterns seen

Line 479-491- This needs to go earlier and be more forceful. You need to account for this limitation.

Line 522- Please fix this sentence. Is Sinovenator representative because it has this small pelvis or it is representative because it is a basal troodontid, that happens to have a small pelvis.

Line 523- A great illustration of this would be a graph of this size in say mm against a body mass estimate (I personally would use femoral circumference but you could do length as well) and show how this taxon is well off the expected.

Line 528- Again maybe at this size related because at about 1-1.5 kg you don't need all that much total mass to run, but you may need to keep the ratios within groups similar to that seen in larger taxa.

Line 534- This is an interesting finding, but then again that pelvis comes from a specimen with an estimated hip height of about 1.8 m per van der Reest and Currie 2017. That’s over twice as big as MOR 748 which is estimated at about 50 kg, so we are talking an animal likely over 100kg and 70-100+ times larger than the basal Sinovenator. It’s likely that the derived troodontid, due to size alone, had different minimum muscle demands compared to the basal one. This could factor in why you suddenly, when you troodontids get back to the size of things like Ornithomimus, you see this value bounce back to similar levels.

Line 578-Or it could mean one was built for long distance versus short burst and prey restraint.

Line 593- Wait, who classified wild jungle fowls as poor runners? Hutchinson and Garcia 2002 call them “adept runners” and Gallus has been know to hit about 4 m/s in a sprint, not bad for an animal that is in the wild less than 2 kg. Now they are more sprinters than long distance runners, but as you have not differentiated those two styles.

Line 612-Or that is the minimum needed to move, even at relatively slow speeds, at large body sizes.

Line 632- Maybe include a couple of less cursorial taxa in this group to be sure of this statement.

Line 638- This is cool, perhaps you should include this in the main section as this would be interesting to readers who don’t want to go into the supporting documents.

---

## Round 0.2 · Minor Revisions

Dear Dr. Rhodes,

Thank you for your submission. Based on reviewer comments, there are a few changes to be made before the manuscript can be put forward. With your revision, please include a response to reviewers rebuttal letter, a tracked changes version of the manuscript, and a clean version of the manuscript.

In particular, I would like you to pay close attention to reviewer 2's comments relating to phylogenetic regressions. There are quite a few trees available now so a PGLS based on a time-calibrated tree should be possible (although I doubt that it will change results much).

In addition to reviewer comments, here are a few specific additional comments from my reading:

Rhodes et al., Maniraptoran Pelvic Musculature

Line 82–83: Can you be a little more specific with regard to which types of anatomical changes? Maybe add (e.g., …) after anatomical changes. Also, do you mean significant evolutionary rate shifts in morphological evolution here?

Line 219: ‘bilaterally symmetric organism’

Line 269: Can you specify which formulae were used here?

Line 479: Grammar

Line 502: While there is some utility in making things simpler, it seems like there is quite a bit of variability in muscle area and the diversity of pelvic architecture in modern birds might mean that the stepwise pattern that has been found in fossil theropods is an oversimplification. Can you add a few sentences to expand on this more in the discussion and conclusion? I think that it is an important point to emphasize.

Line 745: I’m not familiar with this test, but how much of the difference in fits here is actually meaningful? I wouldn’t think they would be statistically distinguishable fits? Is there an equivalent to deltaAICs for model selection with this test?

Line 759: Categorizing the degree of cursoriality, perhaps?
That would make it clearer that you're referring to cursoriality as a spectrum.

Line 764: I would think that the chicken being artificially selected has affected their muscle proportions. Can you discuss that at some point in the paper, probably in the methods?

Line 796: ‘subarctometatarsus’

Line 1496: Do you mean ‘pelves of Varanus’?

Table 7: Please reduce the number of significant digits here.

·

Basic reporting

No comment.

Experimental design

No comment.

Validity of the findings

No comment.

Additional comments

I am satisfied with all of the changes made by the authors, and I am very happy with this manuscript. I have one final request. The panels in Figure 4 that are blurry (A, C, and D) should not be published as is. If suitable photos of these specimens cannot be acquired, then line drawings should be made and used instead. This is critical especially since this is an online journal, and figures will be seen on a screen, and not printed out as tiny figures. There really is no reason to have photographs that are out of focus with anatomical features that are difficult or impossible to see with this being a manuscript about anatomy. Otherwise, this is an excellent manuscript, and I look forward to seeing it published! Well done!

·

Basic reporting

I enjoy reading this article as it is clearly written, appropriately cited and well structured. While there is a lot of data in this manuscript, It does not feel bogged down or a difficult read.

Experimental design

The research question is clear and is addressed in a rigorous manner.

Validity of the findings

There are some areas that need improvement but overall this is a strong paper.

Additional comments

I want to say congratulations, this is a very strong manuscript and I look forward to seeing it published. Overall it is very well done. I also want to thank you all for taking the time to look at all comments from both reviewers adn addressing them clearly where you believe they should be addressed. I enjoyed seeing how you took the suggestions to heart and made the work stronger.

I have a few small issues, mostly they revolve around the allometric section, which I know must be frustrating as I am the one who suggested it in the first place. The big issues are first your input data, especially for extant taxa where actual values are possible, is suspect. For example you list the emu and ostrich at 72 and 248 kg respectively, but this is significantly larger than I have seen reported for even the largest member of either species with typical values of around 31.5-37 for the former and 100-115 in the later (This particular example is taken from Olsen and Turvey 2013). So your values are around twice as high, which of course will influence how your reconstruct them in your regressions. If the specimens you used did not come with a mass then I would suggest using a species average to get a better value. Second, your choice of groups to run is strange because you use all bipeds, and many small groups such as all non-maniraptorans or all maniraptorans, but never all non-avian theropods. Also, if you did want to include birds you should account for phylogenetic relatedness, as it actually does effect your results somewhat. None of these points are even close to being serious, but I think need to be addressed in the next version.

Below are my small comments, listed with the line number per the reviewing copy . If this differs from your version I apologize for the confusion.

Line 299: Here is where I believe you need to address the mass problem as it becomes clear lookint at the data that you values for the larger avians are too high. Also of note, your mass for Gallus specimen is the exact same mass as the Sinovenator, which seems a little too coincidental, and maybe too small. For example Allen et al. 2013 had their Gallus at around 2.5kg and that is not an uncommon mass to see. Just wondering if you had any way of getting a more precise match for the living avians as this can have an effect on your allometry analysis.

Line 360: Maybe “relatively” should be added before small.

Line 361: Replace "That" with "This"

Line 362: Did you look at Mei long? I am sure it won’t change anything but it falls out closer to Sinovenator at the base of the clade in multiple analyses such as Pei et al. 2020. Maybe just confirm it doesn’t alter anything and add it as a good reference if you don’t mind.

Line 588: You did not do a phylogenetically informed regression correct. I just ask because you should expect phylogenetic relatedness to play some role in this value, especially when including extant avians in your dataset.

Line 599: Why didn’t you do all theropods as one permutation? Your sample size is smallish (5 maniraptoran non-avians) so adding in the three non-maniraptorans to get a total of 8 specimens makes the results a bit stronger.

Line 632: This actually depends on what dataset you run. So if you run all non-avian theropods without any phylogenetic correction you get something slightly different. Here the residual for Sinovenator is only -0.041 which is much less than in Allosaurus (-0.096), Albertosaurus (-o.079) or Saurornitholestes (-0.11). When you look at your predicted percentage now Sinovenator is at -3.23%, so not outside the confidence interval. If we do a simple phylogenetically controlled regression (assigning all branches to equal length, which is wrong but it’s not something you want to go down the rabbit hole about how to sort the “real values “ for each branch) we get a predicted percentage of -2.3%. I think the major issue is the fact that Gallus, a modern avian whose different posture, at the exact same mass artificially inflates the differences in Sinovenator. So if you either exclude avians or control for phylogeny this signal goes away. I did this with total muscle area but you see the same thing with extensor area.

Line 851: Maybe change this to "suggests" as A) you only have 2 taxa in your comparison and B) level of crouching may alter this in Sinovenator and C) remember we are reconstructing muscle size % based on a proxy that has a lot of uncertainty thus its hard to be super confident in this.

Line 899: Perhaps make sure you are singling out the eudromaeosaurs as Unenlagiines are very troodontid like.

Line 922: Something you might want to note is if I regress extensor against total muscle area we actually see the ornithomimid has a smaller then expected values. Its not significant but its about the same level (slightly larger actually) than you see in Saurornitholestes.

Line 929: Bates et al. 2009 have a reconstruction, but in the context of the volume of that region and the organism.

---

## Round 0.3 · accepted · Accept

Dear authors,

Thank you for your submission to PeerJ. I appreciate your careful consideration of reviewer comments and am excited to recommend this paper for publication.


On a final reading, I noticed these few lingering issues that should be taken care of prior to publication:

Line 217: Add comma before but

Line 238: remove comma

Line 247: add version number

Line 283: If this was done in excel, remove the part about R since it is not relevant to your study.

Line 290: Capitalize ‘Akaike Information Criteria’ and use ‘Criteria’ rather than criterion in this instance

Line 326: Which package and which version of that package?

Line 589: Grammar

Line 614: delete space before comma


Please let me know if you need any help with your next steps as the article moves to the proof stage.

Best,

Brandon P. Hedrick, Ph.D.